

# Summer Greenland Blocking in observations and in SEAS5.1 seasonal forecasts: robust trend or natural variability?

Johanna Beckmann[1,2,⊣], Giorgia Di Capua[2,⊣], Paolo Davini[3]

[1]School of Earth, Atmosphere and Environment, and ARC Special Research Initiative for Securing Antarctica's Environmental Future, Monash University, Clayton, Kulin Nations, Australia.
[2]Earth System Analysis, Potsdam Institute for Climate Impact Research, Member of the Leibniz Association, Potsdam, Germany
[3]Consiglio Nazionale delle Ricerche, Istituto di Scienze dell'Atmosfera e del Clima, Torino, Italy

⊣shared first authorship

*Correspondence to*: Giorgia Di Capua (dicapua@pik-potsdam.de)

**Abstract.** Given its impact on enhanced melting of the Greenland ice sheet, it is crucial to assess changes in frequency and characteristics of summer Greenland blocking. Indeed, the occurrence of such atmospheric pattern has seen a marked increase in recent decades: however, the observed trend is not captured by any simulation from state-of-the-art global climate models. It is therefore paramount to determine whether the lack of trend is caused by a misrepresentation of key physical mechanisms in climate models or whether such trend is mainly attributable to decadal variability, or both. Here we investigate Greenland blocking characteristics in reanalysis (ERA5) and ECMWF seasonal forecasts (SEAS5.1), showing that about 10% of the 1000 permutations of SEAS5.1 runs can simulate a 43-year trend equal or larger to the ERA5 one: this suggests that the initialization and the higher model resolution contribute to a more realistic representation of the blocking dynamics than in freely-evolving climate runs. To further investigate these aspects, we apply the Peter and Clark momentary conditional independence (PCMCI) algorithm to assess monthly causal pathways. Results show that while the relationship among Arctic temperature, snow cover, Atlantic multidecadal variability and Greenland blocking is consistent both in ERA5 and SEAS5.1, the effect of early snow melt over North America on Greenland blocking is mostly absent in SEAS5.1. Therefore, while it is possible that the observed trend is due to internal decadal variability, the misrepresentation of the snow cover processes may explain the difficulty that SEAS5.1 has in reproducing the observed trend. This deficit in representing the snow impact on the atmospheric circulation might also be the culprit of the missing trend in climate models, raising the question whether long-term projections underestimate a future increase in Greenland blocking and ice melt.



## 1 Introduction

Atmospheric blocking over Greenland can be defined by the presence of a persistent anticyclone in the mid-troposphere in proximity of Greenland. In the boreal summer, it is accompanied by higher than average temperature anomalies at the surface which can significantly contribute to melting of the Greenland ice sheet (Hanna et al., 2014; Tedesco and Fettweis, 2020; Wang and Luo, 2022). The years 2012 and 2019, which rank first and second respectively for negative surface mass balance over the Greenland ice sheet (Tedesco and Fettweis, 2020) encapsulate the essence of such a process. In both years, persistent blocking patterns triggered extreme melting, reaching even the ice sheet's summit. While anthropogenic global warming increases the mean temperature of the troposphere, changes in the frequency and intensity of key atmospheric patterns, such as the Greenland blocking pattern can further enhance melting at the surface (McLeod and Mote, 2016; Wachowicz et al., 2021). Thus, such changes pose an additional threat to the stability of the ice sheet itself and may lead to additional sea level rise (Beckmann and Winkelmann, 2023; Sasgen et al., 2020; Shepherd, Andrew et al., 2021). It is then crucial to establish whether climate models can correctly represent the physical mechanisms that lead to blocking in historical simulations, to gain confidence that future projections do not underestimate the effect of atmospheric blocking on the stability of the Greenland ice sheet. However, climate models - despite having improved their skill in the recent decades (Davini and D'Andrea, 2020) seem to lack agreement with observed blocking trends for summer Greenland blocking, thus raising the questions whether projected future changes for blocking are reliable (Davini and D'Andrea, 2020; Delhasse et al., 2020; Maddison et al., 2024; Masato et al., 2013).

Greenland blocking is a large-scale atmospheric high-pressure low-vorticity system associated with the negative phase of the North Atlantic Oscillation (Woollings and Hoskins, 2008) located over Greenland. Given its large spatial scale of thousands of km, Greenland blocking can have a large impact on atmospheric circulation in the North Atlantic sector. The location of the blocking high can oscillate between south and north Greenland, but it often presents a baroclinic structure, so that the high-pressure at the surface is located eastward of Greenland, close to Iceland (Davini et al., 2012). Greenland blocking is usually generated by the cyclonic wave breaking on the northward flank of the jet stream, which is then followed by retrogression of the ridge toward Greenland (Hauser et al., 2024).

Despite being less studied than its winter counterpart, summer Greenland blocking presents characteristics that can severely impact the Greenland ice sheet (Hanna et al., 2018; McLeod and Mote, 2016; Tedesco and Fettweis, 2020). Due to its anticyclonic structure, during periods of strong Greenland blocking southerly advection of warm air on the western side of Greenland favours enhanced melting (McLeod and Mote, 2016), a phenomenon that has gained relevance in recent years. Indeed, blocking has been associated with extreme melt years in 2010 (Tedesco et al., 2011), 2012 (Nghiem et al., 2012), and 2019 (Tedesco and Fettweis, 2020) and can generally lead to significant summer temperature anomalies (Blau et al., 2024). Moreover, modelling studies show that if these extreme melting years increase in frequency in the future, the contribution to sea level rise from the Greenland Ice Sheet could increase by 20% by the end of the $23^{rd}$ century (Beckmann and Winkelmann, 2023).





Arctic amplification (AA), i.e. the phenomenon that sees the Arctic region warming at a pace two-to-four times faster than the rest of the Northern Hemisphere (Cohen et al., 2014; Davy et al., 2018; Previdi et al., 2021; Serreze et al., 2009) has been

suggested to influence circulation patterns in the northern mid-latitudes (Cohen et al., 2014; Francis and Skific, 2015; Nakamura et al., 2016), however with contrasting results (Blackport and Screen, 2020; Cohen et al., 2020; Screen et al., 2018). While the AA signal is stronger in winter, enhanced Arctic warming in summer can affect weather patterns, leading to weaker storm tracks, shifts in the position of the jet stream and favouring the amplification and persistence of Rossby waves (Coumou et al., 2018). Changes in atmospheric circulation patterns in the North Atlantic sector connected to AA can affect the frequency

and duration of Greenland blocking events (Preece et al., 2023), and in turn increase the effect of the latter on the local ice sheet (McLeod and Mote, 2016). Multiple evidence is accumulating in recent years showing that the frequency of occurrence of summer Greenland blocking has been increasing in the last two decades (Hanna et al., 2018; Wachowicz et al., 2021). However, a clear understanding of the underlying mechanisms leading to such increase is not yet available. Preece et al. (2023) proposed a mechanism which connects reduced snow cover in late spring over the North American continent with the formation

of a Rossby wave pattern, which in turn enhances blocking over Greenland in the following summer. However, the large interannual variability of the region, which is often described as a red noise process (Feldstein, 2000) and can present years of extremely high blocking frequency followed by years where almost no blocking is observed, still questions the real presence of a long-term trend (Davini et al., 2021; Gollan et al., 2015). For example, in recent years the trend of Greenland blocking has weakened (Preece et al., 2023) leaving open the opportunity of being a result of the signal of natural variability. Natural

climate variability modes, such as the Atlantic Multidecadal Variability (AMV), may impact large scale circulation patterns in the North Atlantic, and the local jet stream dynamics, thus indirectly affecting blocking onset and maintenance, and further complicating the scenario (Häkkinen et al., 2011; Luu et al., 2024).

Next to high- and mid-latitude drivers, tropical sea surface temperatures and convective activity in the tropical belt can provide further physical mechanisms that can potentially affect warming trends and blocking features over Greenland and the Arctic

region (Gan et al., 2024; Liu et al., 2016; Matsumura and Kosaka, 2019). In winter, El Niño episodes and anomalous sea surface temperatures in the tropical Pacific can affect surface warming in the Arctic region via a Rossby wave response (Ding et al., 2014; Matsumura and Kosaka, 2019). In summer, a similar mechanism can lead to the formation of Rossby wave trains originating in east-central tropical Pacific, that propagate to the Arctic region enhancing sea ice melting in the Arctic Ocean in September (Baxter et al., 2019). However, central Pacific El Niño events may be responsible for intensified cyclonic

circulation over Greenland in summer and lead to surface cooling (Matsumura et al., 2021). Similarly, Hu et. Al (2016) show that central Pacific El Niño events can strengthen the tropospheric Arctic polar vortex and the circumpolar westerly wind, which can contribute to inhibiting Arctic warming and sea-ice melting in summer. Model experiments forced with tropical sea surface temperatures show that anomalies in mid-to-upper-tropospheric anticyclonic wind may explain half of the observed Greenland surface warming and ice loss acceleration over the last three decades (Topál et al., 2022).

Another element which further tangles up the Greenland blocking puzzle is related to its representation in climate models. Despite modeling blocking has always been a challenge for both numerical weather prediction models and global climate





models, significant improvements have been achieved in the last 25 years, leading to better representation of the patterns, frequency and trends of blocking (Davini and D'Andrea, 2016, 2020). Higher resolution and improved physical mechanisms have lowered the chronic circulation biases that general circulation models reported on the Euro-Atlantic sector and have thus

improved their ability of representing blocking – from a climate point of view – in a fairly good way.  However, evidence is emerging that none of the CMIP5 and CMIP6 models, among the hundreds of ensemble members available, is capable of replicating the observed increasing trend of summer Greenland blocking, and neither project a significant increase until 2100 (Delhasse et al., 2020; Luu et al., 2024; Maddison et al., 2024). Such lack of trends questions the capability of CMIP6 models to capture the signal of the Arctic amplification and significantly weakens the reliability of their climate predictions.

Moreover, CMIP6 models also fail to capture the magnitude of the melting events, such as the one observed in summer 2019, hinting at a potential underestimation of ice sheet mass loss over Greenland (Delhasse et al., 2020). However, if climate models fail to reproduce observed changes (Hanna et al., 2018), then the question whether projected future changes are realistic and reliable remains open. In this context, identifying the atmospheric and surface drivers of Greenland blocking variability, both in observations and general circulation models, can help assess which physical mechanisms govern the variability of Greenland

blocking and which of the identified mechanisms can be confidently reproduced by climate models. Moreover, it is yet unclear whether the observed trend in Greenland blocking is robust across different observational datasets and over different time periods.

Causal discovery can help identifying causal relationships in a set of different climate and atmospheric variables (Di Capua et al., 2020b; McGraw and Barnes, 2020; Runge et al., 2019a), and provides a tool to validate model performance in reproducing

atmospheric causal pathways (Di Capua et al., 2023; Nowack et al., 2020). The Peter and Clark momentary conditional independence (PCMCI) algorithm is a casual discovery tool that helps identifying true causal relationships from spurious correlation values (Runge, 2018; Runge et al., 2019a, b). With respect to other causal discovery tools, such as Granger causality, PCMCI allows us to assess the possible effect of a (set of) third variable(s). However, when compared to other causal discovery techniques, such as the Liang–Kleeman information flow, PCMCI yields very compatible results, thus providing a

robust tool for the identification of causal relationships (Docquier et al., 2024). PCMCI has found several applications in the field of climate and atmospheric sciences, ranging from the identification of tropical – extratropical teleconnections in observations and forecasting models (Di Capua et al., 2020a, 2023), stratospheric pathways leading to sudden stratospheric warming events (Kretschmer et al., 2016, 2018a), climatological variability and forecasting of crop yields, hurricanes and Indian summer monsoon rainfall (Di Capua et al., 2019; Lehmann et al., 2020; Pfleiderer et al., 2020), historical drivers of

compound hot and dry extreme events in central Europe (Tian et al. accepted) to the identification of local and remote atmospheric drivers of circulation variability in the eastern Mediterranean (Di Capua et al., 2024).

In the current work, we aim at investigating the Greenland blocking representation in state-of-the-art multi-member high-resolution monthly-initialized seasonal forecast system, namely the European Centre for Medium-range Weather Forecasts (ECMWF) SEAS5.1 forecast system, to understand to what extent, initialization characteristics and high resolution could help

in capturing the observed summer Greenland blocking trend. Robustness of the climatology of SEAS5.1 has been already



assessed (Davini et al., 2021), so that this will help us disentangling the role of natural variability from that of anthropogenic climate change in affecting the variability and trends of Greenland blocking, hence helping to resolve a piece of the complex puzzle of Greenland blocking trend. The remainder of this manuscript is organized as follow: we first describe the data and method used (Sect. 2), then present the results obtained (Sect. 3) and finally discuss the results (Sect. 4).

## 2 Data and Methods

### 2.1 Data

We analyse daily atmospheric fields and snow cover fields during the boreal summer period (June to August, JJA) using gridded data (0.25°x0.25° spatially averaged to 1°x1°) for the period 1940-2023 from the ERA5 reanalysis dataset (Hersbach et al., 2020) and for the period 1981-2023 from the SEAS5.1 seasonal retrospective forecast dataset (Johnson et al., 2019), both provided by the European Centre for Medium-range Weather Forecasts (ECMWF). Specifically, we consider daily (temporally averaged to obtain monthly samples) geopotential height at 500 hPa (Z500), 2m temperature (T2m), snow cover (Snow), sea surface temperature (SST) and mean sea level pressure (MSLP). Snow is obtained from snow density and snow depth fields (see Text S1 in the Supplementary Material). To focus on dynamic linkages, the interannual variability, seasonal cycle and any long-term trend are removed (unless otherwise specified). For ERA5, we use the entire 1940-2023 period (referred in the text as ERA5-40) to assess long term trends. However, for the most part of the analysis we use the 1981-2023 period (ERA5-81), to allow a direct comparison with the SEAS5.1 dataset. For each of the 3 monthly time steps considered in each JJA season, the trend over the 43 (84) years for ERA5-81 (ERA5-40) is removed and anomalies around zero are calculated, thus removing both the trend and seasonal cycle. Note that data are not detrended when historical trends are analysed.

SEAS5.1 initialized on March 1 (SEAS5.1-03), May 1 (SEAS5.1-05) and June 1 (SEAS5.1-06) have been used. Although SEAS5.1 provides 51 ensemble members, 25 ensemble members each year are publicly available through the Copernicus Climate Data Store, thus a total of 25x43 (1075) model years. SEAS5.1 data are extracted from the initialization date till October 1, thus a total of seven, five and four months are available for SEAS5.1-03, SEAS5.1-05 and SEAS5.1-06 respectively. As for ERA5, also for SEAS5.1 data, we provide monthly averages, remove the trend over the 1981-2023 period and calculate anomalies centred around zero. Linear correlation analysis and probability-trend estimation were conducted for all datasets mentioned. The correlation and causal inference analyses were performed for both SEAS5.1-03 and SEAS5.1-05, except for the causal effect networks (CEN). For causal discovery, only SEAS5.1-03 was used, as the absence of March and April data in SEAS5.1-03 violates PCMCI algorithm requirements (see Section 2.3 for details).





## 2.2 Greenland blocking indices

Atmospheric blocking can be detected in several ways which can lead to apparently contradictory results, strongly affected by the index chosen for their identification (Woollings et al., 2018). As for other blocking regions, multiple approaches have been proposed to identify the blocking anomaly over Greenland (Hanna et al., 2018; Wachowicz et al., 2020). In the current work, we adopt an atmospheric blocking index which follows the original definition by Tibaldi and Molteni (1990) but extended to a 2-D field following Scherrer et al. (2006) and Davini et al (2012a). This is an index based on the reversal of the meridional gradient of the Z500 field: in contrast to other indices based on anomaly measures, it targets the presence of easterly winds to the south of the blocked grid point, providing a dynamical relevant definition of the phenomenon. To define the index, two meridional geopotential height gradients at southern (GHGS) and northern (GHGN) latitudes are defined:

$$GHGS(\lambda_0, \varphi_0) = \frac{Z500(\lambda_0, \varphi_0) - Z500(\lambda_0, \varphi_S)}{\varphi_0 - \varphi_S}, \tag{1}$$

$$GHGN(\lambda_0, \varphi_0) = \frac{Z500(\lambda_0, \varphi_N) - Z500(\lambda_0, \varphi_0)}{\varphi_N - \varphi_0}, \tag{2}$$

where $\varphi_0$ ranges from 30° to 75°N and $\lambda_0$ ranges from 0° to 360°E. $\varphi_N = \varphi_0 + 15$ and $\varphi_S = \varphi_0 - 15$, such that the gradients are always calculated 15° North and 15° South of $\varphi_0$. The GHGS index defines where the reversal is happening, and it is the key element for the identification of blocking. GHGN ensures the presence of westerly winds northward to the blocked area, focusing on the presence of a high-pressure dome over the analysed region. To ensure that the blocking index is not detecting a poleward migrating subtropical ridge – a condition typically occurring in summer months in Northern Hemisphere (i.e. that easterly winds are not caused by trade winds), a supplementary condition from Davini et al. (2012a) is introduced, and named $GHGS_2$:

$$GHGS_2(\lambda_0, \varphi_0) = \frac{Z500(\lambda_0, \varphi_s) - Z500(\lambda_0, \varphi_{S_2})}{\varphi_S - \varphi_{S_2}}, \tag{3}$$

Where $\varphi_{S_2 =} \varphi_s - 15°$.

Consequently, at daily time scales, blocking is thus identified when:

$$GHGS(\lambda_0, \varphi_0) > 0,$$

$$GHGN(\lambda_0, \varphi_0) < -\frac{10m}{°lat}$$

and $GHGS_2(\lambda_0, \varphi_0) = -\frac{5m}{°lat}$



Blocking is thus a binary index, which can be spatially averaged over Greenland (55°W-0°, 67°N-75°N) to produce what we define as the Greenland blocking index (GBI).

However, such a binary index focuses on the exceptional non-linearity of blocking itself. Given that the target of this analysis is the long-term trend in the strength and shape of geopotential height over Greenland, we also introduce a continuous (non-binary) index, which is obtained as the linear combination of GHGS and GHGN. Indeed, at high latitudes, GHGS$_2$ is found to have a negligible role (not shown). Thus, we define the Greenland gradient index (GGI) so that $GGI = GHGS - GHGN$, which is then highly correlated with GBI. GGI represents an alternative measure to standard blocking and can be physically interpreted as a measure of the strength of the geopotential height anomaly. In both cases – given the limited impact that such constraint has on the real pattern and frequency of blocking – we followed the approach of previous literature (Davini and D'Andrea, 2020; Tibaldi and Molteni, 1990) and we do not apply any spatial or temporal constraint to the blocking index.

## 2.3 Causal discovery and casual inference

In the present work, we apply the Peter and Clark momentary conditional independence (PCMCI) algorithm, a causal discovery tool which allows to disentangle spurious from actual *causal* relationships in a set of uni-variate time series. In general, (cross-) correlation measures are often used to determine the existence of concurrent behaviour between pairs of time series. However, correlation does not imply causation, and spurious correlations can arise from common effects like (i) the presence of strong auto-correlation in one (or both) time series, (ii) common drivers or (iii) indirect links. Thus, two time series may be both driven by a third time series (common driver) and therefore show a significant correlation even though no real causal link is present. PCMCI deals with these issues by iteratively testing for the presence of significant partial correlations (in the linear framework) between pairs of time series by conditioning on a set of one (or more) further time series.

The PCMCI algorithm is composed of two steps, the PC-step and the MCI-step. In the PC-step, the algorithm first identifies all significant lagged (up to a certain prescribed lag $t_{max}$) correlations among all possible combinations of time series in the set of time series (actors), e.g. $P = \{A, B, C, D, F\}$. The first set of "potential parents" for each time series is identified, e.g. with $t_{max} = -3$, $P_A^0 = \{A_{t=-1}, B_{t=-2}, C_{t=-1}, D_{t=-3}, D_{t=-2}\}$. Then, the algorithm tests whether the partial correlation between $A_{t=0}$ and each element $x_i$ in $P_A^0$ is still significant when conditioning on a combination of the remaining elements $x_j \neq x_i$ in $P_A^0$. The partial correlation is calculated by (linearly) regressing $A_{t=0}$ and $x_i$ on $x_j$, and then taking the correlation of the residuals of $A_{t=0}$ and $x_i$. If this partial correlation is still significant, then $A_{t=0}$ and $x_i$, will be considered "conditionally dependent", meaning that their correlation cannot be explained by the linear combination of $x_j$. In the PC-step, the number of conditions $x_j$ gradually increases until the number of elements in the set of potential parents $P_A^n$ is equal or smaller than the number of iterations. At the end of the PC-step, each time series in $P$ will have its own set of potential patents, e.g. $P_A^n = \{A_{t=-1}, C_{t=-1}, D_{t=-2}\}$ and $P_B^m = \{B_{t=-2}, D_{t=-1}, A_{t=-2}, E_{t=-3}\}$.

In the MCI-step, the partial correlation is calculated again between all potential pairs of actors in $P$. However, this time the partial correlation is only calculated once by conditioning on the common set of parents identified in the PC-step. E.g. if $A_{t=0}$



and $B_{t=-1}$ are tested, then the partial correlation will be calculated using the joint set of parents of $A_{t=0}$ and $B_{t=-1}$, the latter being equal to the set of parents of $B_{t=0}$ which has been adjusted for lag -1. The actors found to be conditionally dependent in the MCI-step define the set of causal parents for each time series. Finally, the significance of the detected causal links (each partial correlation has its own $p$-value) is corrected by applying the Benjamini-Hochberg false discovery rate (Benjamini and

Hochberg, 1995) , which accounts for the fact that the same hypotheses are iteratively tested multiple times. The corrected $p$-values then determine whether a link between two actors is considered "causal".

The causal effect is then calculated by linearly regressing each actor on its own set of causal parents. E.g., assuming that the causal parents of $A_{t=0}$ are $P_A^n = \{A_{t=-1}, C_{t=-1}, D_{t=-2}\}$, then the following multilinear regression equation will be calculated:

$$A_{t=0} = \beta_A * A_{t=-1} + \beta_C * C_{t=-1} + \beta_D * D_{t=-2} + \xi_A \tag{4}$$

Where $\beta_i$ is the causal effect of each parent $x_i$. With standardized actors, a $\beta_B$ value of 0.5 corresponds to a change of 0.5 s.d. of $A_{t=0}$ given a change of 1 s.d. of $B_{t=-1}$ (given that all other parents remain unaltered). In this work, we use the PCMCI algorithm as coded in Tigramite, version 5.2, and we use the PCMCIplus function, which allows to calculate links also at lag 0. However, lag 0 links are not "directed", which means that the direction of causality is not determined. The Tigramite package is publicly available and can be found at this link https://github.com/jakobrunge/tigramite.

The causal relationships obtained with PCMCI are then plotted in a causal effect network (CEN). In a CEN, each actor is represented by a node in the network, while lagged causal relationships are represented by directed arrows, showing the direction of causality. The colour of the arrows (nodes) shows the strength of the causal effect, the $\beta$ value (auto $\beta$ value), while the numbers on the arrows show the lag at which the causal link is detected.

Finally, the causal effect can also be calculated by applying the concept of causal inference, rather than causal discovery. When

causal inference is applied, the causal links are determined a priori (generally based on expert-knowledge of the analysed physical system) and then the causal effect is calculated following the same multi-linear regression model described earlier in Eq. (4). A detailed description of the Tigramite functions and parameters used can be found in Text S2 in the Supplementary Material.

## 2.3 Bootstrapping and significance

To ensure statistical significance in the seasonal forecast data (SEAS5.1 initialized in March, May, and June), we employed bootstrapping. We resampled the original set of 25 ensemble members x 43 years, creating 10,000 new samples by randomly selecting one of the 25 ensemble members for each year along the timeseries from 1981 to 2023. This resampled distribution of 10,000 timeseries is illustrated in various figures. When a single value for SEAS5.1 is mentioned, it refers to the median of the bootstrapped distribution. For example, the correlation between two indices in SEAS5.1 is determined by calculating the

correlation values for all 10,000 timeseries and then taking the median of these values.



## 3 Results

### 3.1 Greenland blocking indices and trends

Greenland blocking in the summer season has increased its frequency in recent decades. Figure 1a and 1b show the grid-point blocking frequency trends defined by the JJA yearly averaged blocking index for ECMWF ERA5 reanalysis for two different time windows. However, notable differences emerge when focusing on different time windows: while in ERA5-40 the blocking trend is positive in most of the Northern Hemisphere (Fig. 1a), the linear trend shows much larger variability when ERA5-81 is considered (Fig 1b), showing decrease in western Europe and western North America. Still, over the Greenland Blocking

region, highlighted by the red box in both panels, trends remain positive irrespectively of the period considered.

The difference between the two trends may be related to the large interannual variability which characterizes blocking. Blocking can easily achieve a frequency of 20% in certain summers and then show an almost negligible frequency in the following year. This can be also seen in Figure 1c, where both the GBI (blue) and GGI (red) indices are reported, and the year-to-year variability emerges evidently. A significant positive trend is observed in both cases (trends are significant at 99%

confidence level for both indices with a Mann-Kendall test). However, recent years are characterized by a reduction of blocking from 2015 onward, which weakens the magnitude and the significance of the trend (Fig. S1 in the Supplementary Material). It is important to notice that the yearly averaged GGI and GBI indices are also highly correlated, showing a Pearson correlation coefficient of $r = 0.8$. From a statistical point of view, this implies that the two indices are very similar and in the rest of the manuscript we will employ the GGI to perform PCMCI analysis, which works best with no-binary indices.







**Figure 1: Greenland blocking index observed trends.** Panel (a) shows blocking trend (shading) and blocking climatology (green contours in boreal summer (JJA) over the Northern Hemisphere for ERA5-40. Contours are drawn every 3% blocked days. Panel (b): same as for Panel (a) but for ERA5-81. Panel (c) shows JJA Greenland Blocking Index (blue) and Greenland Gradient Index (red), measured as the averaged over the red box shown in panel (a) for ERA5-81. Dashed lines show the season average, bold lines the 10-year running mean and the thin solid lines the linear trend. Values for the trend and their p-values (estimated with a Mann-Kendall test) are shown in the legend for both indices.





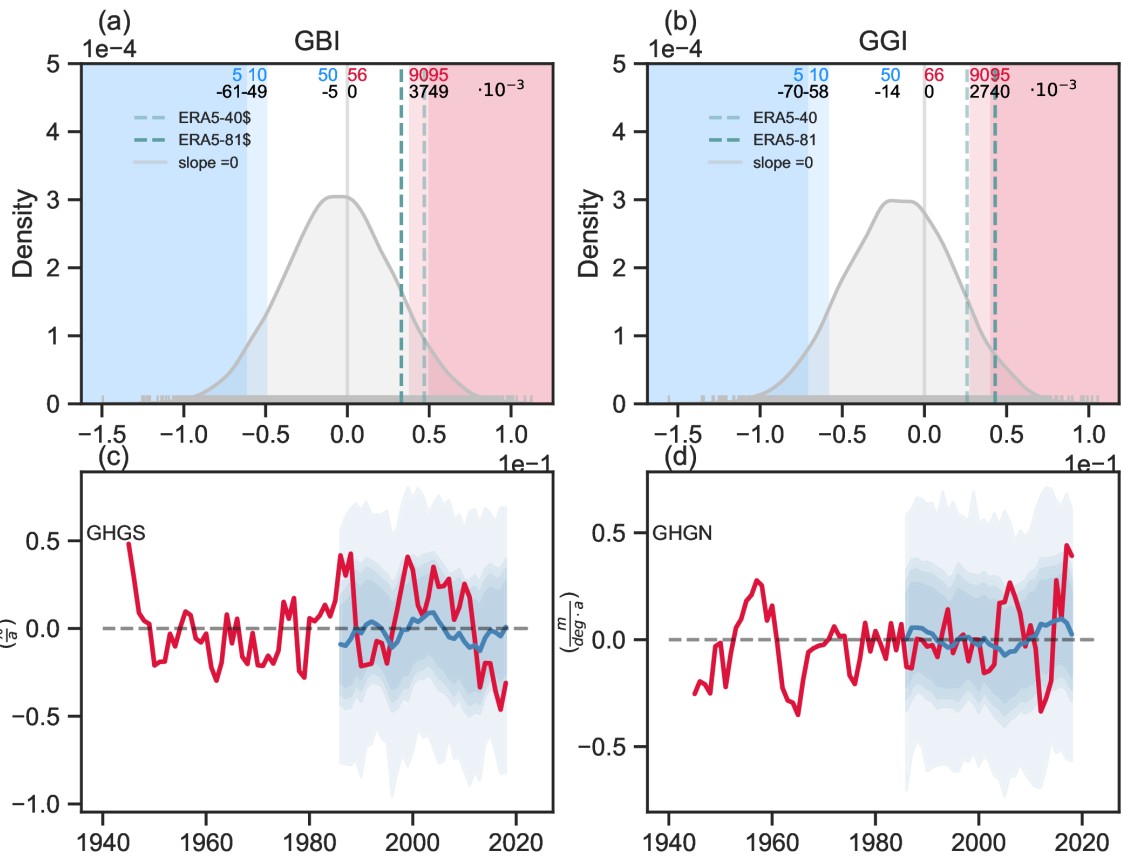

**Figure 2: Greenland blocking and Greenland Gradient index trends in SEAS5.1-03 and ERA5.** Panel (a) shows the Probability density function of JJA trends in GBI for the $10^4$ different member combinations. Panel (b): same as for panel (a) but for the GGI index. Shaded vertical lines show values 5th,10th, 90th and 95th percentiles. Percentiles are shown in blue (5th,10th, 50th) and red (90th and 95th) with the corresponding trend values in black below. Percentile of the distribution of a slope of 0 is also given in red and indicated with the grey vertical line. Green dashed verticals indicate linear slope of ERA5-40 (light green) and ERA5-81 (solid green). Panel (c) shows the 11-year running trend of ERA5 (red) and the SEAS5.1-03 trend distribution (blue shadings) for the GBI. Panel (d) same as for panel (c) but for the GGI index. The dark blue line indicates the median 11-year running mean of the SEAS5.1-03 distribution.

Observed trends in GBI and GGI are shown in detail in Fig. 2. To compare the ERA5 trend with SEAS5.1-03 trends, we calculate a probability density function (PDF) of the $10^4$ trends derived from our bootstrapped SEAS5.1-03 ensemble members (see Methods Section 2.4). In contrast to ERA5, the median values of the SEAS5.1-03 PDFs indicate negative trends for both GGI and GBI (Figs. 2a and 2b respectively). The ERA5-81 trend falls right below the 90th percentile for GBI (Fig. 3a) and between the 90th and the 95th percentiles for GGI (Fig. 3b). This suggests that albeit SEAS5.1-03 can reproduce the observed trends with certain specific combinations of ensemble members, the probability of doing so is relatively low (~10-5%). Using



later initialization dates (SEAS5.1-05 and SEAS5.1-06) does not increase the likelihood of matching the observed trends (see Fig. S2 in the Supplementary Material). We provide the same analysis using the GHGS and GHGN components of the Greenland Blocking index (Fig. S3 in the Supplementary Material) and show that the probability of reproducing the observed

trend is slightly higher for GHGN, while it is very low for GHGS (below 2.5%), indicating that GHGS may be the primary source of the low agreement between ERA5 and SEAS5.1 trends.

To investigate the multidecadal variability of the blocking trends, we run an 11-year running trend on the ERA5 and SEAS5.1 timeseries (Figs. 2c and 2d). SEAS5.1 trends are shown both as the ensemble mean trend (blue solid line in Fig. 2c-d), or as the distribution of trends obtained from each of the $10^4$ realizations (blue shading in Fig. 2c-d). The timeseries of ERA5 reveals

that the short-term blocking trend (for both GBI and GGI) fluctuates between positive and negative values, indicating a strong influence from climate variability on decadal timescale. Particularly in recent years, the running mean trend shows remarkable negative trends, which explains the loss of significance for the GBI observed for the ERA5-81 trend (Fig. S1). The ERA5 running mean trend lies in the ensemble of the bootstrapped SEAS5.1 ensemble, where the median SEAS5.1 trend shows a switching of the sign as well but not always aligned with ERA5.

Next, we check the relationship between GGI and surface temperatures over the extended North Atlantic sector (Figs. 3a-b), to quantify the relationship between high GGI indices and positive temperature anomalies over the Greenland ice sheet. We calculate composites of T2m and Z500 fields during days for which GGI values are greater than their JJA s.d. (GGI > 1 s.d.) for both ERA5-81 (Fig. 3a) and SEAS5.1-03 (Fig. 3b) datasets. Z500 composites for ERA5 show a ridge centred over eastern Greenland, which is paired with a trough to the south, centred west of the British Isles (Fig. 3a). This north-south dipole pattern,

which resembles a negative North Atlantic Oscillation (NAO) phase (though shifted northwards), is also present with a similar shape, location and strength of the Z500 anomalies in the SEAS5.1-03 composite (Fig. 3b). Positive T2m anomalies are shown over Greenland for both ERA5 and SEAS5.1-03, reaching values of +2.5K while over the Arctic Ocean and the North Atlantic weak negative anomalies are shown (-.0.5K, shading in Figs. 3a-b). The PDF of the GGI index is shown in Fig. 3c for ERA5-81 and Fig. 3d for SEAS5.1-03 and provides a measure of the daily variability of the Z500 field over Greenland in summer.

The JJA GGI mean value is -1.52 m for ERA5-81 and -1.39 m for SEAS5.1-03, while the s.d. values are 10.1 m and 11.8 m respectively (Figs. 3c-d). Thus, the variance in SEAS5.1-03 is slightly overestimated with respect to ERA5-81. Note that GGI values are detrended and the anomalies are centred around zero (as described in Section 2.1).

High pressure systems over Greenland are associated with high temperature anomalies both in SEAS5.1 and ERA5 (Figs 3a-b). Temperature over Greenland is then calculated by spatially averaging T2m values over the box 67-81°N,25-55°W, and the

corresponding index is named T2m-Greenland (T2m-G). The JJA climatological PDFs for T2m-G for ERA5 and SEAS5.1-03 are shown in Figs. 3e and 3f respectively (orange shading in Figs. 3e-f). T2m-Greenland has a mean value of 265.7 K (264.8 K) for ERA5-81 (SEAS5.1-03) and a s.d. of 2.1 K (3.1 K) for ERA5-81 (SEAS5.1-03). The effect of GGI on T2m is then quantified by calculating the T2m-G PDFs sub-selecting only days for which GGI > 1 s.d. (shown are red lines in Figs. 3e-f). For ERA5-81, the T2m-G PDF strongly shifts towards higher T2m values for GGI > 1 s.d., with 85% of the values falling





above the 50th quantile and 28% of the data exceed the climatological 90th quantile, thus showing a 3-fold increase in the probability of extreme heat (Fig. 3e). Similarly, the SEAS5.1-03 T2m-G PDF for days where GGI > 1 s.d. also shows a strong shift towards higher T2m values, with 88% of the data falling above the 50th climatological quantile and 33% of the data exceed the 90th climatological quantile, consistently showing a 3-fold increase in extreme T2m-G values (Fig. 3f). Hence, high GGI values are generally related to higher than average T2m-G values both in ERA5-81 and SEAS5.1-03. However, the relationship

to extreme T2m-G values is more marked in SEAS5.1-03. The same Figures but for GGI SEAS5.1-05, and for SEAS5.1-03 GHGS and GHGN show consistent results, though for positive GHGN values the sign of the Z500 and T2m anomalies is inverted (see Supplementary Material, Figs. S4, S5 and S6 respectively).



**Figure 3. GBI and extreme temperatures over Greenland.** Panel (a): composite of JJA Z500 (contours) and T2m (shaded) anomalies for
(daily) time steps with GGI > 1 s.d. for ERA5-81. Panel (b): Same as panel (a) but for the SEAS5.1-03 dataset (25 ensemble members each



year). Panel (c): probability density function (PDF) for the Greenland gradient index (GGI) daily climatology for ERA5-81. Panel (d): Same as for Panel (c) but for SEAS5.1-03 dataset. Panel (e): PDF for the T2m-Greenland (T2m-G) daily climatology (orange) and sub-selected days with GGI > 1 s.d. (red) for ERA5-81. Panel (f): Same as for Panel (2) but for SEAS5.1-03.

### 3.2 Potential drivers of Greenland blocking at monthly time scales

We identify four monthly potential drivers of GGI based on hypothesized mechanisms described in the Introduction Section. These potential drivers are: (i) T2m over the Arctic circle (T2m-Arctic, an indicator for Arctic amplification), (ii) snow cover over northern North America (Snow-NAm), (iii) MSLP over the eastern US (MSLP-NAm, following the mechanism identified by Preece et al 2023) and (iv) the Atlantic multidecadal variability (AMV) index. The spatial domain and acronym of each index is reported in Table 1. Figures 4a,e show the composites of T2m fields over months where the T2m-Arctic index is above

its own 1 s.d. (T2m-Arctic > 1 s.d.) for ERA5-81 and SEAS5.1-03 respectively. As expected, high surface temperature anomalies up to 2°C are seen over the polar cap (latitude > 60°N) over land, while over the Arctic Ocean weak negative anomalies are detected. This result confirms the tendency of high-latitude land to warm up more strongly in summer, compared to the Arctic Ocean (Di Capua et al., 2021). Over mid-latitude and tropical land-ocean surfaces, T2m anomalies tend to be positive both in ERA5-81 and SEAS5.1-03. However, in SEAS5.1-03 T2m anomalies are much weaker outside the Arctic

circle (Fig. 4e), while in ERA5 strong warming is also detected over Europe, western Russia, the Middle East, southwestern US and mid-latitude North Pacific (Fig. 4e). This discrepancy between ERA5 and SEAS5.1-03 may be explained by the different size of the two sub-samples: while ERA5-81 contains a total of 3*43 = 129 months is summer, SEAS5.1-03 contains 25 times more. Since 1940, T2m-Arctic has shown a significant positive trend that has doubled in the time frame from 1981-2023 (Figure S7 in the Supplementary Material). The 11-year running mean (Fig. 4i) shows that T2m-Arctic fluctuates between

positive and negative values since 1940 but has stayed positive since around 2000. The majority of the bootstrapped SEAS5.1-03 experiences this continued positive trend as well indicated by the blue-shaded ensemble distribution in Fig. 4i and the SEAS5.1-03 median values in thick blue, showing that the polar cap warming is well captured by the model.

| Index | Acronym | Spatial domain |
|---|---|---|
| Greenland blocking index | **GBI** | 55°W-0°, 67°N-75°N |
| Greenland gradient index | **GGI** | 55°W-0°, 67°N-75°N |
| Geopotential height gradient South | **GHGS** | 55°W-0°, 67°N-75°N |
| Geopotential height gradient North | **GHGN** | 55°W-0°, 67°N-75°N |
| Atlantic multidecadal variability | **AMV** | 80°W-0°, 0°-60°N |
| Snow cover over North America | **Snow-NAm** | 70°-135°W, 40°-75°N |
| Mean sea level pressure over North America | **MSLP-NAm** | 70°-105°E, 30°-50°N |
| Artic 2m temperature | **T2m-Arctic** | 180°W-180°E, 60°-90°N |

**Table1. Summary of indices with their corresponding acronyms and spatial domains.**



Composites for Snow over North America during months with Snow-NAm < 1 s.d. (May only) are shown in Fig. 4b,f for ERA5-81 and SEAS5.1-03 respectively. ERA5-81 shows lower than average Snow anomalies over Alaska, at the border between the US and Canada, and in Siberia (Fig. 4b), while SEAS5.1-03 only shows negative Snow anomalies over North America (Fig. 4f). Since 1960, the 11-year running mean trend of Snow (Fig. 4j) has generally shown negative values, with only brief and minor episodes of positive values. In recent years, there has been a positive trend in the ERA5 Snow data;

however, the overall snow cover remains below the 1940-2023 mean (Fig. S7 in the Supplementary Material). Meanwhile, the SEAS5.1-03 ensemble exhibits significant fluctuations between positive and negative 11-year running mean trends (Fig. 4j), suggesting that snow cover may not be accurately captured in SEAS5.1.

Composites for MSLP fields during months with MSLP-NAm < 1 s.d. are shown in Figs. 4c and 4g for ERA5-81 and SEAS5.1-03 respectively. Months with low MSLP-NAm are associated with low MSLP anomalies over the US, Canada, the mid-latitude

North Atlantic both for ERA5-81 and SEAS5.1-03. High MSLP anomalies are shown over the Arctic circle and Greenland (Figs. 4c-g). The 11-year running mean trend of MSLP-NAm has shown strong fluctuations between positive and negative trends throughout the entire period from 1940 to 2023 (Fig. 4k). These fluctuations are well captured by the bootstrapped SEAS5.1-03 ensemble since 1981. Over the whole timeseries, we can see a statistically significant negative trend in MSLP-Nam for ERA5, with a doubling of the negative slope since 1981 (Fig. S7 in the Supplementary Material).

Finally, composites for SST anomalies during months with AMV > 1 s.d. are shown in Fig. 4d,h for ERA5-81 and SEAS5.1-03 respectively (the AVM is calculated following Zhang et al (2019), see Text S3 in Supplementary). Positive SST anomalies are found over high-latitude North Atlantic, west of the European and African coasts and over the tropical North Atlantic, while cold anomalies are shown over mid-latitude eastern North Atlantic (Figs. 4d,h). Cold anomalies are also seen over the western North Pacific for both ERA5-81 and SEAS5.1-03, while warm anomalies over mid-latitude North Pacific are clearly

seen only in SEAS5.1-03 (Fig. 4h). The AMV exhibits strong fluctuations in both the sign and values of the 11-year running mean trend, which are well represented by the bootstrapped SEAS5.1-03 ensemble (Fig. 4l).





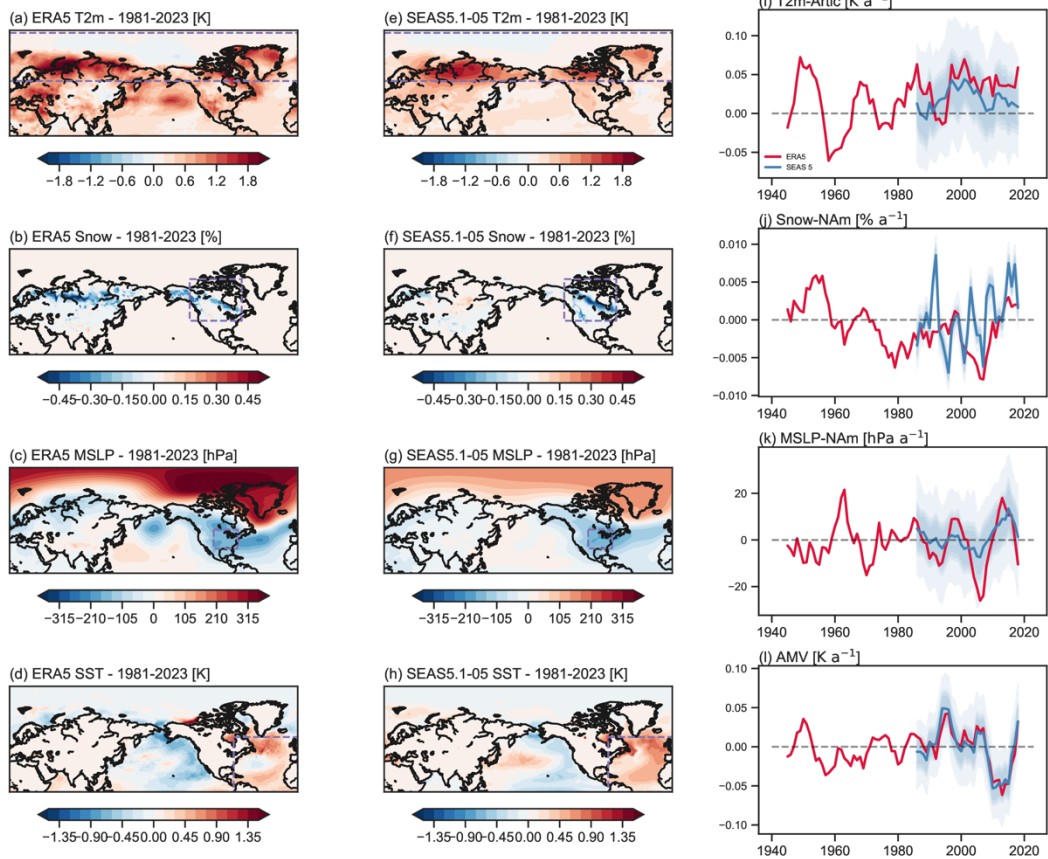

**Figure 4. Composites and 11-year running mean of monthly indices.** Panel (a) shows the composites of JJA T2m fields for months with
T2m-Arctic > 1 s.d. for ERA5-81. Panels (b) same as for Panel (a) but for May Snow fields during months with Snow-NAm < 1 s.d.. Panel
(c) as for Panel (a) but for MSLP fields during months with MSLP-NAm < 1 s.d.. Panel (d) as for Panel (a) but for SST fields during months
with AMV > 1 s.d.. Panels (e)-(h) same as for panels (a)-(d) but for SEAS5.1-03.The grey dashed box in (a-h) indicates the area from which
the monthly indices (T2m-Arctic, Snow-NAm, MSLP-NAm, AMV) were derived. Panels (i) illustrates the 11-year running mean trend of
T2m-Arctic for ERA5 (red) and the SEAS5.1-03 distribution of $10^4$ timeseries in blue shades. Panels (j)-(l) same as for panel (i) but for
Snow-NAm, MSLP-NAm, and AMV. The thick blue line indicates the median value of the total distribution. The grey dashed line indicates
the running mean trend of ~0.



## 3.3 Relationship among climate drivers and blocking indices

One of the goals of this study is to test the Preece et al. (2023) hypothesis, hereafter referred to as "Preece23", which suggests that Arctic amplification (AA) may induce early snow cover depletion in May over North America, generating a low-pressure

anomaly, which in turn could act as a waveguide, leading to blocking over Greenland. Additionally, we aim to assess whether summer Greenland blocking is influenced by the AMV and evaluate SEAS5.1's ability to replicate the observed correlations. To investigate the relationship between various blocking indices (GBI, GGI, GHGS, GHGN) and key climate drivers (AMV, T2M-Arctic, Snow-NAm, MSLP-NAm), we calculate a correlation heatmap using detrended variables from ERA5-40, ERA5-81, and SEAS5.1-03 (Fig. 5). All time series are linearly detrended, except for ERA5-40 T2M-Arctic, which is detrended using

a polynomial function due to its non-linear trend.

We first focus on the GBI index in relation to Preece23's hypothesis. In ERA5-81, positive (lag 0) correlations are found between both T2M-Arctic, MSLP-NAm and GBI (r ~ 0.3), while Snow-NAm (taken in May) shows a negative correlation with GBI (r ~ -0.2). The correlation between AMV and GBI is r ~ 0. These results support the first part of the hypothesis regarding the relationship between AA, decreased snow cover and enhanced GBI, but not the relationship between a low-

pressure anomaly over North America and increased GBI (Fig. 5b). However, only the correlation between T2M-Arctic and GBI is statistically significant ($p < 0.05$).

When comparing the original blocking index (GBI) to the derived GGI, the correlation values are qualitatively similar with some differences: the correlation between MSLP-NAm, T2m-Arctic and GGI decrease and are not significant for ERA5-81, while the correlation with Snow-NAm increases to r ~ 0.35 and is significant (Fig. 5b). The correlation between GHGS and

the precursors indices is qualitatively similar to that of GGI, with the difference that GHGS also shows a negative (r ~ -0.3) and significant correlation with MSLP-NAm. Figure S5 reveals that GHGS>0 leads to the gradients reversal with a high-pressure system over Greenland, which could favour blocking. However, the fact that this negative correlation does not show up for GBI and GBI may be because the correlation between GHGN and MSLP-NAm is also negative (r ~ -0.5). GBI requires both GHGS and GHGN conditions to be met simultaneously, while GGI is a linear combination of the two. Notably, GHGN

must be negative to contribute to blocking. Thus, negative MSLP anomalies can lead to positive pressure anomalies over Greenland, which promote blocking (GHGS>0), but also negative pressure anomalies over eastern Greenland (GHGN>0), reducing the likelihood of blocking. As a result, the effects cancel out in GGI, leading to near-zero correlations with MSLP-NAm. Thus, disentangling the blocking indices into GHGS and GHGN reveals more details of the complex relationship between MSLP-NAm and Greenland blocking.


Extending the analysis to ERA5-40 generally results in weaker and less significant correlations, although qualitatively the sign of the correlation is consistent (Fig. 5a). Notably, the correlation between Snow-NAm, MSLP-NAm and GHGS and between Snow-NAm and GHGN are significant (r ~ -0.2-0.3, Fig. 5a). This suggests that the proposed Preece23 mechanism may not hold consistently over the longer historical period. Two interpretations are possible: (i) Arctic amplification may have become



more dominant only in recent decades and (ii) the stronger signal observed in ERA5-81 may result from sampling of internal variability.

When the correlation between the potential drivers among each other is considered, both ERA-40 and ERA-81 generally show consistent qualitative results (Figs. 5a and 5b respectively). Both AMV and T2m-Arctic exhibit negative correlations with MSLP-NAm and Snow-NAm (r ~ -0.1-0.4), while T2m-Arctic and AMV are positively correlated (r ~ 0.3). Snow-NAm and

MSLP-NAm are also positively correlated (r ~ 0.1-0.4). However, the strength of these correlations varies between the periods, with stronger correlations found in ERA5-81 (Fig. 5b). In general, no direct correlation is found between AMV and the blocking indices, indicating that AMV's influence on blocking may be indirect, primarily acting through MSLP-NAm.

Finally, we assess whether SEAS5.1-03 can replicate the observed correlations (Fig. 5c). Due to the large number of bootstrapped time series, SEAS5.1-03 shows lower signal-to-noise ratios, leading to generally weaker correlations. The

correlation values in Fig. 5c represents the median of the distribution for each pair of time series, with standard deviations shown in Fig. S8 in the Supplementary Material. Despite this difference, SEAS5.1-03 replicates key patterns, especially the correlations among AMV, T2M-Arctic, and Snow-NAm. The positive correlation between T2m-Arctic and the blocking indices is weaker (r ~ 0.1-0.2) but consistent with what is shown in ERA5-81. In contrast, SEAS5.1-03 struggles to capture correlations between Snow-NAm, MSLP-NAm, and the blocking indices, except for a weak negative correlation between

MSLP-NAm and GHGS (r ~ -0.2). As snow cover in SEAS5.1-03 is derived from climate model variables rather than observational data, we test whether a later initialization in SEAS5.1 (closer to observed snow cover) could improve these correlations (see Fig. S8 in the Supplementary Material). We examine SEAS5.1-05, which uses ERA5.1 snow cover initialized on May 1, and SEAS5.1-06, which allows for snow cover over the entire month of May as calculated in ERA5-81. While this later initialization strengthens correlations among the climate drivers, it does not improve their relationship with the blocking

indices.

Thus, in ERA5 our findings generally support the first part of Preece23's hypothesis, showing that T2M-Arctic and May snow cover may influence MSLP over North America, which in turn favours pressure highs over Greenland (GHGS>0). However, this does not consistently lead to blocking, as MSLP also contributes to GHGN>0, reducing the likelihood of blocking. The AMV appears to have an influence on MSLP and Arctic temperature, potentially affecting blocking indirectly. While

SEAS5.1-03 to -06 captures relationships among climate drivers similar to those seen in observational data, they fail to replicate the full extent of snow cover's role in blocking indices, particularly in relation to MSLP. This suggests a potential misrepresentation of the driver-blocking interactions in the model. Further analysis in the next Section will explore causal relationships between these drivers and blocking indices.



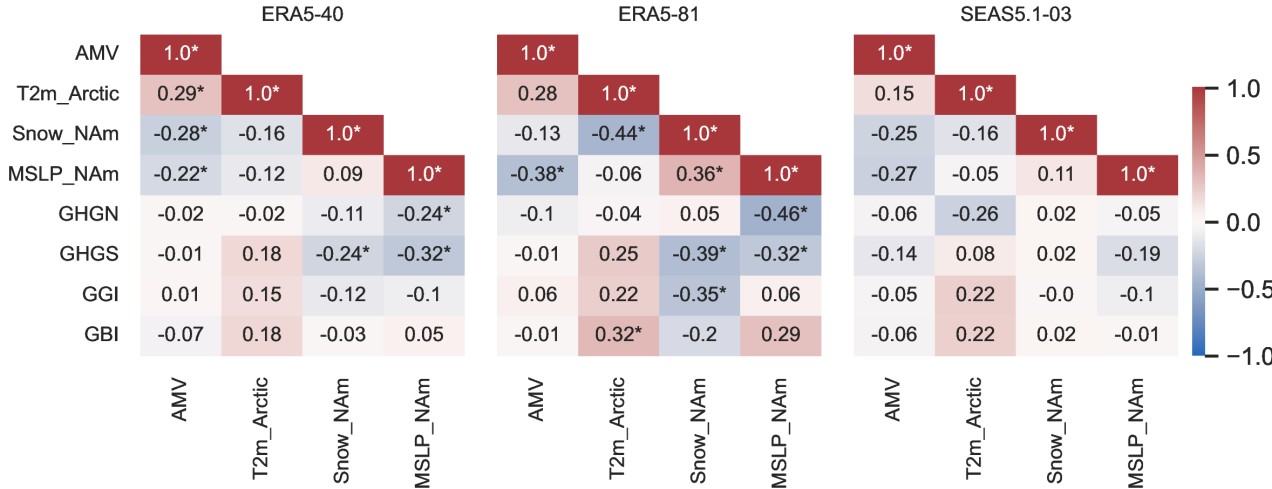

**Figure 5. Summer correlation heat maps.** Panel (a) shows the correlation of the different blocking indices (GBI, GGI, GHGS, GHGN) and the four identified potential monthly drivers (AMV, T2M-Arctic, Snow-NAm, MSLP-NAm) in ERA5-40 for the summer season. Panel (b) same as for panel (a) but for ERA5-81. Panel (c) same for panel (b) but for SEAS5.1-03. All data were linearly detrended except for T2M-Arctic in ERA5-40 which was detrended using a polynomial fit. Statistically significant values at $p < 0.05$ are marked with an asterisk. The SEAS5.1-03 correlation values represent the median correlations from the $10^4$ bootstrapped samples. The correlations were analysed for summer (JJA averaged) for all variables, while snow cover data were taken from the monthly mean in May.

## 3.4 Causal relationship in ERA5 and SEAS5.1

To disentangle spurious correlations from causal relationships, we now apply PCMCI to detect the causal links between GGI and its potential drivers. First, we apply PCMCI to build a causal effect network (CEN) for monthly GGI, Snow-NAm, T2m-Arctic, MSLP-NAm and AMV. We use $\text{lag}_{max} = -2$ for ERA5-81 and $\text{lag}_{max} = -1$ for SEAS5.1-03 (this is necessary not to violate PCMCI requirements, see Methods Section 2.3), while $\text{lag}_{min}$ is set to 0 for both ERA5-81 and SEAS5.1-03. Note that, given the shortness of the time series, we do not apply the FDR correction to ERA5-81, which would result in too few significant causal links. Thus, we use the obtained CENs not to make final statements about causal relationships, but rather to get a first reasonable estimate of the potential causal links and to provide a first comparison between ERA5-81 and SEAS5.1-03. The ERA5-81 CEN (Fig. 6a) shows two incoming causal links towards GGI at lag –2, one negative ($\beta = -0.15$) from Snow-NAm and one positive ($\beta = 0.15$) from T2m-Arctic. A negative undirected link (lag 0) between AMV and GGI is also present ($\beta = -0.15$). These links mean that lower than average Snow over North America during AMJ is followed by higher than average GGI values in JJA. Similarly, though with opposite signs, higher than average T2m over the Arctic circle in AMJ are followed by higher than average GGI values in JJA. Thus, these links would support Preece23's hypothesis that early snow





cover depletion over North America can lead to more blocking. Similarly, the link from T2m-Arctic hints to the potential relevance of Arctic amplification for blocking. The undirected link between AMV and GGI shows that lower than average AMV values in JJA correspond to higher than average GGI values in the same months. MSLP-NAm shows only one incoming positive ($\beta$ = 0.15) link from Snow-NAm at lag –2, meaning that higher than average Snow-NAm values in AMJ are followed

by higher MSLP anomalies over western US, also in keep with Preece23. Snow-NAm shows three incoming links: one positive link from MSLP-NAm at lag –1 ($\beta$ = 0.15), and two negative links, one from T2m-Arctic at lag –1 ($\beta$ = -0.3) and one from AMV at lag –2 from AMV ($\beta$ = -0.15). Thus, higher than average Arctic temperatures and SST over the North Atlantic are both followed by lower than average Snow anomalies in the following months, while higher than average MSLP-NAm anomalies lead to higher Snow anomalies. Thus, Arctic warming has the potential to contribute to snow depletion and in turn

to increased blocking. T2m-Arctic shows only one incoming negative link at lag –2 ($\beta$ = 0.15) from Snow-NAm. Finally, AMV also shows only one positive incoming link at lag –1 from T2m-Arctic ($\beta$ = 0.15).

The same CEN but for SEAS5.1-03 is shown in Fig. 6b. In general, a similar network structure is shown, with the following differences: the GGI is connected to T2m-Arctic via a positive undirected link (lag 0, $\beta$ = 0.25), to MSLP-NAm via a negative undirected link (lag 0, $\beta$ = -0.15) and to AMV by a negative undirected link (lag 0, $\beta$ = -0.1). A direct link from Snow-NAm

is not present in SEAS5.1-03. The AMV shows one additional undirected link with MSLP-NAm (lag 0, $\beta$ = -0.1) and two negative incoming links at lag –1 from MSLP-NAm ($\beta$ = -0.1) and GGI ($\beta$ = -0.1). Two undirected links are also shown between MSLP-NAm and Snow (lag 0, $\beta$ = 0.1) and between the latter and T2m-Arctic (lag 0, $\beta$ = -0.1). These results show that ERA5-81 and SEAS5.1-03 generally agree on the causal links found among MSLP-NAm, Snow-NAm and T2m-Arctic. However, the direct causal link from Snow-NAm towards GGI is missing, while the undirected link from MSLP-NAm appears.

Note that ERA5-81 and SEAS5.1-03 CEN are not one-to-one comparable, because (i) the maximum lag used is different, (ii) the length of the two time series is different and (iii) a different set of causal precursors will by construction lead to different $\beta$-values.

In general, similar causal links are also detected when GHGS and GHGN are analyzed (Fig. S9 in Supplementary Material). For ERA5-81, most links are consistent with what shown in Fig. 6a, with the following differences: (i) AMV and Snow-NAm

are linked by a two-way negative causal link (Figs. S9a,b), (ii) the link from T2m-Arctic towards GHGS is missing (Fig. S9a) and (iii) MSLP-NAm shows a direct link towards both GHGS and GHGN (Figs. S9a,b). For SEAS5.1-03, again most links are consistent with what shown in Fig. 6b, with the following differences: (i) no direct link is found between MSLP-NAm and GHGN (Fig. S9d) and (ii) GHGS shows two additionally outgoing links towards T2m-Arctic and AMV (Fig. S9c). Note that GHGN links should have an opposite sign with respect to those shown for GHGS in order to have the same effect on GGI.

E.g. a positive (negative) influence of T2m-Arctic on GHGS (GHGN) has a joint positive effect on GGI. In contrast, the negative effect of MSLP-NAm on both GHGS and GHGN (Figs. S9a-b) cancels out when GGI is considered (Fig. 6a) meaning we do not find the proposed connection between the wave source (MLSP-NAm) and the blocking (GGI, Fig. 6a) suggested by Preece23. However, given the direct link from snow cover to GHGS, but not to GHGN, this relationship extends to GGI, supporting the hypothesis that early snow cover depletion in May might influence blocking patterns. Although the direct



association between the AMV and the blocking gradients is not observed, an indirect influence is suggested through a connection with MSLP-NAm, indicating at least some level of natural variability affecting the blocking indices GHGS and GHGN.

Thus, in general, the proposed chain of causality supports the idea that all four potential drivers influence, directly or indirectly, blocking over Greenland. In keep with Preece23, higher Arctic temperatures are linked to snow cover depletion, which is in
turn linked to negative MSLP anomalies over North America. While it is unclear whether these negative anomalies can lead to blocking, their influence on the meridional gradients is apparent. A direct link from snow cover towards GGI is only found in ERA5-81, while both model and reanalysis show a link to Arctic temperatures. The Atlantic multidecadal sea surface temperature variability is also negatively affecting blocking over Greenland and indirectly GHGS and GHGN, supporting the hypothesis that natural variability also plays a role.


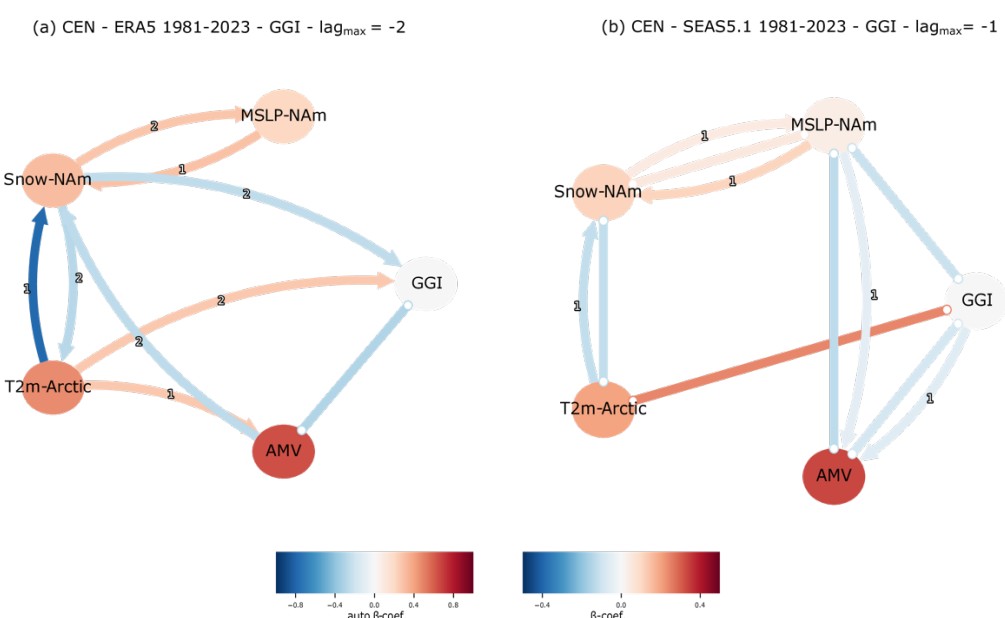

**Figure 6. Causal effect networks for ERA5 and SEAS5.1-03.** Panel (a): CEN with MSLP-NAm, Snow-NAm, T2m-Arctic, AMV and GGI for ERA5-81 and $\alpha = 0.05$ (no FDR applied, see Methods Section 2.3). Panel (b): same as for Panel (a) but for SEAS5.1-03 and with
FDR correction applied.

To provide a fair comparison between ERA5-81 and SEAS5.1-03, we adopt the concept of causal inference (see Methods Section 2.3). Based on the results shown in Fig. 6, we now assume that the selection of causal links shown in Fig. 7a is found



in both ERA5-81 and SEAS5.1-03, and also for GHGS and GHGN indices. Figures 7b-e show the $\beta$ values calculated for
ERA5-81 and ERA5-40, together with the $\beta$ value distributions obtained by bootstrapping 1000 times SEAS5.1-03 and
SEAS.1-05. For links that do not include GGI (Fig. 7b), in general ERA5-81 and ERA5-40 $\beta$ values fall inside the SEAS5.1-
03 and SEAS.1-05 $\beta$ value distributions. The AMV $\rightarrow$ MSLP-NAm link (lag 0) show negative $\beta \sim$ -0.1-0.2, with SEAS5.1
showing somewhat higher absolute values up to $\beta \sim$ -0.2 (Fig. 7b). The T2m-Arctic $\rightarrow$ Snow-NAm link (lag -1) shows the
strongest causal effect with a $\beta \sim$ -0.3-0.5, however, both ERA5 values are outside or at the far end of the two SEAS5.1
distributions, which are cantered around $\beta \sim$ -0.1-0.2. This hints to a potential underestimation of the effect of MJJ Snow on
JJA MSLP over North America in SEAS5.1. In contrast, the Snow-NAm $\rightarrow$ T2m-Arctic link (lag -1) is cantered around $\beta \sim$ 0 in
both ERA5 and SEAS5.1, showing a negligible effect of snow cover on North America on Arctic surface temperatures. Both
the Snow-NAm $\rightarrow$ MSLP-NAm link (lag -1) and the MSLP-NAm $\rightarrow$ Snow-NAm link (lag -1) show a weak positive $\beta \sim$ 0.1-0.2 on
each other, with the latter being somewhat stronger in SEAS5.1.

Figures 7c, 7d and 7e show the causal links directed towards GGI, GHGS and GHGN respectively. For GGI, three links show
good agreement between ERA5 and SEAS5.1, i.e. AMV $\rightarrow$ GGI (lag 0, $\beta \sim$ -0.1), T2m-Arctic $\rightarrow$ GGI (lag 0, $\beta \sim$ 0.2-0.4) and GGI
$\rightarrow$ GGI itself (lag −1, $\beta \sim$ 0.1-0.2). For these links, ERA5 $\beta$ values fall inside the distribution of SEAS5.1 $\beta$ values. This is not
the case for links T2m-Arctic $\rightarrow$ GGI (lag -1, ERA5 $\beta \sim$ 0.3, SEAS5.1 $\beta \sim$ 0.1), Snow-NAm $\rightarrow$ GGI (lag -1, ERA5 $\beta \sim$ -0.2, SEAS5.1
$\beta \sim$ 0) and MSLP-NAm $\rightarrow$ GGI (lag 0, ERA5 $\beta \sim$ 0.1, SEAS5.1 $\beta \sim$ -0.2). For the latter, the (absolute) $\beta$ values are higher in
SEAS5.1 than in ERA5, while SEAS5.1 struggles to reproduce the strength of the causal effect of Snow-NAm and Arctic-T2m
on GGI (Fig. 7c). GHGS (Fig. 7d) and GHGN (Fig. 7e) show similar results. However, when the two components that make GGI
are analysed separately, it is possible to distinguish whether the underestimation effect in SEAS5.1 comes from the southern
or norther components of the blocking gradients. E.g., the underestimation of the GGI $\rightarrow$ Snow (lag −1) link comes from an
underestimation of the GHGS $\rightarrow$ Snow causal link (Fig. 7d), while the underestimation of the GGI $\rightarrow$ T2m-Arctic (lag −1) link
comes from an underestimation of the GHGN $\rightarrow$ T2m-Arctic link (Fig. 7e).

In general, for 7 out of the 11 causal links analyses there is a good match between ERA5 and SEAS5.1. However, SEAS5.1
seems to underestimate the links between both Arctic surface temperatures and snow cover over North America and blocking
indices (Fig. 7c-e). Moreover, the influence of snow cover on MSLP over North America is also underestimated (Fig. 7b).
Thus, the causal analysis supports the hypothesis that the forecast model does not correctly represent the physical effects of
depleted snow cover on increased blocking over Greenland.



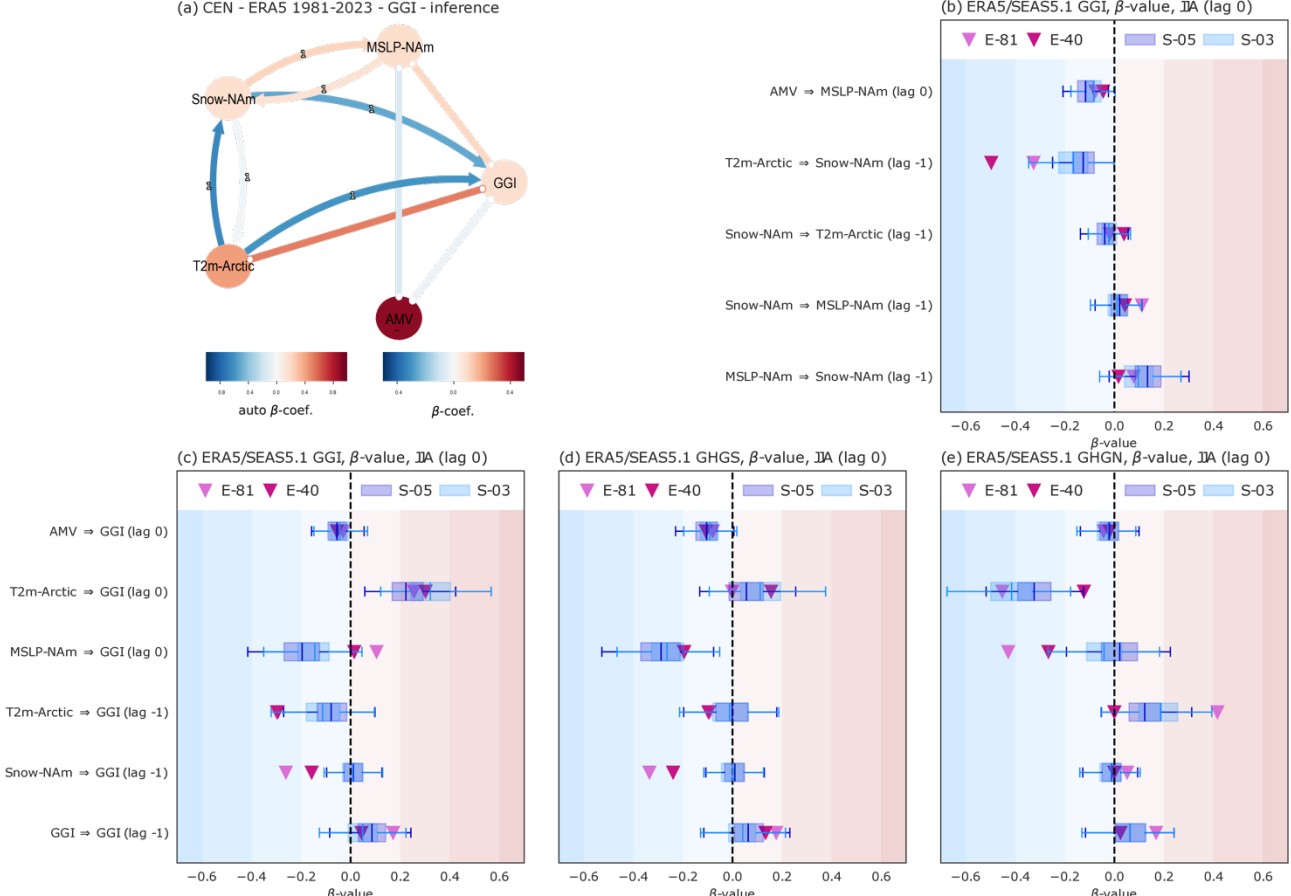

**Figure 7. Causal inference for monthly potential precursors.** Panel (a): inferred CEN with ERA5 MSLP-NAm, Snow-NAm, T2m-Arctic, AMV and GGI for the 1981-2023 period for both SEAS5.1-03, SEAS5.1-05, ERA5-40 and ERA5-81. Panel (b): $\beta$ values for links shown in Panel (a) that do not include GGI. Panel (c): same as for Panel (b) but for links that are directed towards GGI. Panel (e) and (f): same as for Panel (c) but for GHGS and GHGN indices. The boxplots show $\beta$ values calculated a 1000 time in SEAS5.1-03 and SEAS5.1-05 (whiskers show the 95% confidence interval).

## 4 Discussion and Conclusion

In this study, we investigate whether the observed trend in summer Greenland blocking can be attributed to natural variability, specifically the Atlantic multidecadal variability (AMV), or whether it is influenced by Arctic amplification (AA), following the hypothesis proposed by Preece23. According to Preece23, AA may lead to early snow cover depletion in May, which can generate a low-pressure anomaly over North America, which subsequently may act as a waveguide, resulting in increased blocking over Greenland.

On top of this, our study aims to address a critical gap in understanding the representation of Greenland blocking within advanced seasonal forecasting systems by evaluating the ECMWF SEAS5.1. Following the definition of (Davini et al., 2012)



for the Greenland blocking index (GBI), we derived a Greenland gradient index (GGI) from its southern and northern meridional components, GHGS and GHGN, to ensure a continuous time series for statistical analysis where the use of the GBI was not applicable. Both blocking indices (GBI, GGI) show a positive statistically significant trend from 1940 to 2023 but reveal a negative trend in the last decade. The analysis focuses on how SEAS5.1 state-of-the-art, multi-member high-resolution model captures the spatial patterns, trends, and natural variability associated with Greenland blocking.

Using various statistical methods (linear regression, correlation analysis, causal discovery, and causal inference), we find multiple evidence that both Arctic amplification and natural variability have influenced Greenland blocking at monthly time scales during the observational period. Our correlation and causal analyses indicate a robust relationship in ERA5 over the period 1981-2023, where high Arctic temperatures lead to early snow cover depletion in May, which can subsequently cause a low-pressure anomaly over North America (MSLP-NAm), consistent with Preece23 (Figs. 5b, 6a and 7b). While in ERA5

the direct effect of MSLP-Nam on GGI seems limited (Fig. 7c), its effect on GHGS is consistent with Preece23: negative MSLP-NAm lead to positive GHGS anomalies (Fig. 7d). However, negative MSLP-NAm also lead to positive GHGN anomalies (Fig. 7e), likely explaining the lack of signal on GGI. Snow cover influences MSLP-NAm (Figs. 6a and 7b), and shows a direct effect on GGI and GHGS, but not on GHGN. Enhanced Arctic temperatures also positively influence GGI and GHGS, while GHGN receives a consistent negative influence (Figs. 7c-e). Thus, our analysis reveals that in ERA5, higher

Arctic temperatures and reduced snow cover lead to low-pressure anomalies over North America but do not result in blocking, as defined by the index used in our study. However, we observe a connection with elevated pressure highs over Greenland (GHGS > 0). This finding aligns with Preece et al. (2023), where the authors apply the blocking index proposed by Hanna et al., which defines blocking as positive pressure anomalies over Greenland (Hanna et al., 2013). Notably, this index lacks some characteristics typically associated with blocking. The degree of similarity between this index and GHGS > 0, as well as

whether it sufficiently captures Greenland blocking, remains an open question. Addressing this gap could provide valuable insights and warrants further investigation in future studies.

While SEAS5.1 struggles to reproduce the observed blocking trend (only ~10% of the sub-samples show a trend that is comparable with the ERA5 trend, Fig. 2), our analysis sheds light on the ability of the forecasting model in reproducing the physical mechanisms observed in ERA5. SEAS5.1 shows consistent causal links among high Arctic temperatures, low snow

cover, and low-pressure over North America (Figs. 6b and 7b), which agree both with ERA5 and with the Preece23 hypothesis, although the strength of these connections is generally underestimated in the model, suggesting a low signal-to-noise ratio. However, the influence of snow cover on the blocking indices is the most misrepresented causal link (Figs. 7c-e). Thus, this missed mechanism may help explaining the difficulties of SEAS5.1 to replicate the historical trends in ERA5. A potential explanation for this missed link may be that snow cover itself is misrepresented is SEAS5.1: May snow cover climatology in

SEAS5.1 shows an overestimation over western North America both in SEAS5.1-03 and SEAS.1-05 (see Supplementary Material Figs. S10o and S11o respectively). This hypothesis is also supported by the strong differences in the 11-year running mean between ERA5 and SEAS5.1 (Fig. 4j). While SEAS5.1 data initialized in March and May shows very similar causal results (Fig. 7), using SEAS5.1 initialised in June, improves at least partially the correlations among Arctic temperatures, snow





cover, and low-pressure over North America (Fig. S8). However, it does not improve the correlations with the blocking indices,
suggesting that even using snow cover values closer to the observations does not improve the misrepresentation of the snow –
blocking link in SEAS5.1.

While blocking shows multidecadal variability in its observed blocking trends, the influence of the Atlantic multidecadal
variability (AMV) on blocking variability is only weakly detected. The 11-year running mean trends in the blocking indices
(Fig. 2) and their potential drivers (Fig. 4) indicate strong variability over the observational period. The AMV shows a
negligible causal influence on blocking indices (Fig. 7c-e). Nevertheless, AMV has at least some influence on the pressure-
low and snow cover over North America (though the latter is found only on ERA5, Fig. 6), and thus has an indirect, though
weak, influence over blocking. The AMV shifted to a positive trend in 1981, likely enhancing the negative trend of MSLP
and, in turn, blocking trends. Thus, our analysis shows that the combination of high Arctic warming and a positive AMV phase
likely favour enhanced blocking over Greenland. However, in the past six years, we observed a weakening in the blocking
trend (Fig. 2), despite a steady increase in Arctic temperature and decrease in low-pressure over North America (Figs. 4j and
4k respectively), indicating that internal variability might be overshadowing the AA signal. In fact, in that same period a small
positive trend in the Snow cover of North America was observed, which could explain the dampening of the blocking. If this
mechanism holds, we could further speculate in the short-term that the positive phases of the AMV will enhance the probability
of low pressure over North America, thereby increasing both GHGS and GHGN. However, if snow cover will continue to
decrease, positive values of GHGS may prevail, favouring larger blocking frequencies.

Blocking is important for Greenland surface temperatures and Greenland melting events (Hanna et al., 2018; McLeod and
Mote, 2016; Sasgen et al., 2020). Thus, the uncertainty in how climate models can reproduce blocking variability and trends
(Delhasse et al., 2020; Maddison et al., 2024) represents a challenge to correctly estimate the future changes that will affect
the Greenland ice sheet, and as a consequence, the impact of the latter on sea level rise (Beckmann and Winkelmann, 2023).
Our analysis builds confidence that the ECMWF seasonal forecasting system can reproduce, though rarely, the observed trend,
highlighting the limitation of CMIP6 where none of the models is able to replicate the trend. Moreover, the seasonal forecast
model can represent most of the Arctic – blocking links, with the exception of the direct link of snow cover on blocking. Thus,
while blocking over Greenland does not show increasing trends in future projections (Davini and D'Andrea, 2020), we show
that there is at least a chance that the lack of this trend is explained by suboptimal representation of the snow – blocking
mechanisms in the model.

Future changes in the causal drivers of blocking can then provide us with an estimation of the potential effect that these changes
may have on blocking itself. While our analysis shows that positive phases of the AMV can enhance blocking activity over
Greenland, the AMV variability is projected to decrease in future projections from CMIP6 models (Shang et al., 2024).
Moreover, AMV variability is also too weak in CMIP5 and CMIP6 models (Bracegirdle, 2022), thus potentially affecting its
effect on blocking. Snow cover extent over North America will keep declining (O'Gorman, 2014; Quante et al., 2021) while
Arctic temperatures are projected to keep increasing under anthropogenic climate change (Cai et al., 2021; Hu et al., 2021;
Shu et al., 2022). This combined effect has the potential to contribute to increased blocking over Greenland and then accelerate





melting of its ice sheet. However, the amount of snowfall in winter is probably influenced by large-scale oceanic and atmospheric patterns (Lundquist et al., 2023) meaning that climate variability will continue to influence blocking. Nonetheless,

long-term future Arctic temperatures could shift snowfall to rainfall  (Collins et al., 2024; McCrystall et al., 2021). Thus, while blocking will still be modulated by the AMV or other sources of natural variability, in the long term, snow cover will decline due to the dominant forces of climate change, potentially leading to increased blocking.

While future projections show a steady decline of mean snow cover over northern latitudes (O'Gorman, 2015; Diro and Sushama, 2020;  Mudryk et al., 2020, Quante et al., 2021) observations show an increase in snow cover over the past decade

(Cohen et al., 2020). This trend may be explained by changes in several mid-latitude and polar drivers, such as meandering of tropospheric circulation (Di Capua and Coumou, 2016; Francis and Vavrus, 2015), Arctic Amplification (Cohen et al., 2020) and polar vortex variability (Cellitti et al., 2006; Kretschmer et al., 2018a,b ; Overland and Wang, 2019). On the other hand, tropical – extratropical (Arctic) teleconnection can affect the circulation patterns and climate variability in the Arctic regions (Meehl et al., 2018a; Ye and Jung, 2019). Convective activity variability of both tropical Pacific and Atlantic Oceans are

partially responsible for negative trends in Arctic sea ice extent in the past decades (Meehl et al., 2018a) and tropical Indo-Pacific convection can induce a poleward-propagating Rossby wave train, which in turn affects surface temperature over the Arctic Archipelago during summer (Zhu et al., 2024). Thus, changes in tropical regions, or changes in these teleconnection pathways, can potentially enhance the effect of climate change in Arctic regions.

To summarize, in this study, we investigated the role of snow cover over North America, Arctic amplification and Atlantic

multidecadal variability in influencing Greenland blocking trends. Our findings suggest that higher Arctic temperatures lead to early snow cover depletion, causing low-pressure anomalies over North America, which may increase blocking over Greenland, in alignment with the theory proposed by Preece et al (2023).

Moreover, this work looks at different components of Greenland blocking, thus identifying the specific effect of each causal driver on both its southern and northern meridional components. The ECMWF SEAS5.1 model struggles to accurately

represent the causal links between Arctic temperatures, snow cover, and surface pressure patterns, potentially limiting its ability to reproduce observed blocking trends. Indeed, SEAS5.1 can reproduce the Arctic temperature – snow cover – low pressure links while struggling in reproducing the direct causal link between snow cover and Greenland blocking, hinting to a potential missing physical mechanism that may explain the discrepancy between observed and modelled trends. The model misrepresentation of other phenomena at play, like the AMV variability, which shows at least some indirect influence on

blocking, and tropical drivers, which are not directly considered in this study, may also provide further source for uncertainty. Furthermore, a negative blocking trend in the last decade challenges the idea that Arctic amplification is the main driver of changes in atmospheric blocking over Greenland. Thus, the recent decline in the observed blocking trend, despite stable Arctic temperature increases, indicates that other factors may be at play. Despite the above-mentioned limitations, SEAS5.1 can reproduce the observed Greenland blocking trend in only 10% of its samples, but does not rule entirely the possibility that

natural variability is playing a considerable role in the current observed trend of summer Greenland blocking. This of course opens the question to what extent such linkage might be affecting other regions in the high latitudes and offsetting other long-



term predictions in global climate models. Future work is needed to further assess the capability of historical and future projections provided by CMIP6 models in representing this snow – blocking mechanisms, to evaluate the reliability of future trends in blocking over Greenland.


**Data availability.** The data used in this article can be accessed by contacting the corresponding author. ERA5 (https://doi.org/10.24381/cds.50ed0a73, Copernicus Climate Change Service, Climate Data Store, 2018) and SEAS5.1 (https://doi.org/10.24381/cds.adbb2d47, Hersbach et al., 2023) datasets are publicly available on the Copernicus website.

**Supplement.** The supplement related to this article is available.

**Author contributions.** J.B., G.D.C. and P.D. equally contributed to the design of the analysis. J.B. and G.D.C. performed the analysis and wrote the first draft of the paper. J.B., G.D.C. and P.D. contributed to the interpretation of the results and to the writing of the paper.


**Competing interests.** The contact author has declared that none of the authors has any competing interests.

**Acknowledgements**. We thank Efi Rousi for the valuable discussions at the onset of this research project.

**Financial support.** This research has been supported by the German Federal Ministry for Education and Research (BMBF) via the JPI Climate/JPI Oceans project ROADMAP (grant no. 01LP2002B) (G.D.C.) and via the ClimXtreme project (subproject PERSEVERE, phase 2nd, grant no. 01LP2322D) (G.D.C.). J.B. was further supported by the Australian Research Council Special Research Initiative Securing Antarctica's Environmental Future (SR200100005).

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
