# Peer review of "Summer Greenland Blocking in reanalysis and in SEAS5.1 seasonal"

_EGUsphere, 2024_

## Author Comment (AC1)

**Final response to Anonymous Referee #1**

*We thank the Anonymous reviewer #1 for their insightful and encouraging review. Below we explain in detail how we intend to address the reviewer's comments during the revision process. Our replies are highlighted in italics.*

**Major comments**

1. Observations and reanalysis are repeatedly conflated in the paper. Reanalysis is still a model-derived product, and its snow cover is biased (e.g. Mudryk et al 2015, Mortimer et al 2020) when comparing to in-situ and observation-derived gridded products. This makes me wonder how dependent the results in this paper are on the use of ERA5 as 'observations', and I'd recommend first that the authors are more careful about their use of the word observations, and second that some discussion around how ERA5's biases could be impacting the results. I also wonder whether other reanalysis products would be able to reproduce the same causality? Or whether a different metric for GB would yield similar results, both for causality and for how unusual the reanalysis trend is. I wonder as well where there is a state-dependence and how that might come in to play, for example a non-linearity when future snow cover over North America is much lower on average?

*We agree that – despite being a common terminology in climate modelling - using the word "observations" when reanalysis data are used can generate confusion. Therefore, we will refer to "reanalysis" only in the revised version of the manuscript. Following the reviewer's suggestions, we intend to check whether a different GB metric yields to the same results. Specifically, we plan to use the GBI index defined as the area averaged 500 hPa geopotential height within the domain of 60-80°N and 20-80°W., as described in Preece et al. (2023). We might expect some difference in trends, with the reversal index being strong, but the overall findings should be similar (Luu et al. 2024). Moreover, to address potential state dependence of our analysis, we plan to check whether the causal effect networks change during years with high or low snow cover over North America. This technique has been used in the past in Di Capua et al. (2023) and Tian et al. (2024).*

*Tian, Y., Giaquinto, D., Di Capua, G. et al.: Historical changes in the Causal Effect Networks of compound hot and dry extremes in central Europe. Commun Earth Environ 5, 764 (2024). https://doi.org/10.1038/s43247-024-01934-2*

*Di Capua, G., Coumou, D., van den Hurk, B., Weisheimer, A., Turner, A. G., and Donner, R. V.: Validation of boreal summer tropical–extratropical causal links in seasonal forecasts, Weather Clim. Dynam., 4, 701–723, https://doi.org/10.5194/wcd-4-701-2023, 2023.*

*Luu, L.N., Hanna, E., de Alwis Pitts, D. et al. Greenland summer blocking characteristics: an evaluation of a high-resolution multi-model ensemble. Clim Dyn 62, 10503–10523 (2024). https://doi.org/10.1007/s00382-024-07453-2*

2. Despite the title, quite a lot more time is spent on the idea that there is a forced positive trend in GB, driven by the Preece et al 2023 mechanism, rather than the idea that natural variability (in particular anything other than the AMV), or even a forced increase in variability, has caused the trend in reanalysis. Evidence from CMIP6 is that the forced trend is negative with a lot of variability super-imposed, and so even if the Preece et al 2023 mechanism is correct and is missing from models, it's not obvious to me that that means the models are wrong in the direction of their trend. Perhaps the forced trend for GB is not driven from the pole, but rather from the lower latitudes (on balance) and that's the source of the decline in future GB? I do agree, however, that a missing mechanism that increases GB variability on an interannual timescales could still be important for

future Greenland melt, and I do think that the results here are useful science, I'm just not sure about the way it has been framed.

*We agree with the anonymous reviewer that, while our analysis suggests that the snow cover mechanisms is missing in SEAS5.1, and potentially also in CMIP models – which are not part of this study -, this does not rule out that other mechanisms are at play and that (natural) tropical variability could be influencing the observed trend. While we do not directly focus on tropical forcings in this manuscript, we will address this comment in the revised version of the manuscript, to better highlight in the discussion section that while we show that the snow cover mechanism is important for GBI variability and is too weak/missing from seasonal forecasts, climate models showing a GBI decrease in future projections could still be correct if natural variability, or other mechanisms not addressed here, prevail over the snow cover mechanism.*

3. The intro and the conclusions are both long and meandering at times between forcing of GB between the tropics, midlatitudes and poles, and between climate models and observations. Please consider re-writing to make it clearer.

*Following the reviewer's comment, we plan to shorten and simplify both the introduction and discussion sections, to improve the readability and clarity of the manuscript.*

4. I don't think using T2m-Arctic as an indicator for Arctic amplification is sufficient. A difference between the Arctic and some mid-latitude band would probably be better, as a year with high T2m Arctic could also have high temperatures in general, i.e. T2m Arctic is highly correlated with T2m global. In general, I think the term Arctic amplification is used when the authors intend to say Arctic warming, so I'd recommend more careful wording.

*We agree with the reviewer's suggestion and, in the revised version of the manuscript, we plan to redefine our Arctic amplification index as the ratio between T2m Arctic and Global T2m.*

**Minor comments**

L143: Is the mean of each month for the entire period removed from that month? Following sentence is obvious and need not be included.

*We will remove this sentence following the reviewer's suggestion.*

L150 Why isn't April one of the initialisations for SEAS5.1?

*We chose to analyse both 1. March and 1. May initialisation dates for SEAS5.1 to assess the dependency of the results on the length of the forecasts prior to the target season, e.g on the forecast lead time. It would be reasonable to assume that forecasts initialized on 1. April, may show a mixed signal of both the earlier and later initialised runs. However, since we do not detect significant differences between SEAS5.1-03 and SEAS5.1-05, it is reasonable to assume that an initialisation date in between would not diverge from the obtained results.*

L155: Everything after 'Linear correlation should be moved to the section 2.2

*Lines 155-158 will be moved to section 2.2 in the revised version of the manuscript.*

Figure 2: It's interesting that there's a reversal in the positions of ERA-40 and ERA-81 in terms of their percentile between GBI and GGI. The red lines do not look to be correlated in (c) and (d), as in Figure 1(c). Is there is a mistake in the plot or in the caption? Why is GHGS and GHGN written on panels (c) & (d)?

*We thank the reviewer for point to this mistake. We will correct the figure text in the revised version of the manuscript.*

Figure 3: I wonder if a difference plot of (b)-(a) would be helpful for visualising where ERA5 and SEAS5.1 differ

*In the revised version on the manuscript, following the reviewer's suggestions, we will provide the additional panels showing the difference between ERA and SEAS5.1-03 for both T2m and Z500 fields.*

Figure 4: (j) It's interesting that all the members are so tightly constrained for Snow-Nam compared to other fields, and I wonder why that might be, and if it's showing a related issues, whereby the seasonal model is not simulating variability in snow cover properly?

*Being the seasonal forecast simulations initialised for a state as much close as possible to observations, they tend to diverge with increasing forecast lead time. However, fields such as sea surface temperature (SST) or snow cover are characterized by larger inertia and have a slower variability than atmospheric fields such as T2m or Z500, so it is absolutely expected that they diverge from the initialization state more slowly. Therefore, both AMV (which is derived from SST) and snow cover time series in Figs. 4i and 4l show smaller spread around the average values when compared to T2m and MSLP fields. Moreover, for snow cover (Fig. 4i), only May is considered, and because these plots are obtained using SEAS5.1-05, the divergence from the initial state is minimal. Therefore, this behaviour should not highlight an underlying issue, but rather an expected behaviour of the seasonal forecast fields. We will make sure that this point is clearly explained in the revised version of the manuscript. (double-check whether variability increases for SEAS5.1-03, figure should be in the SI)*

Paragraph L396: non-significant correlations can't support a relationship, the only thing that's been shown there is that Arctic temp and GB are correlated.

*While we think worthwhile to report non-significant correlations (given that the significance is also affected by the shortness of the time series), we agree with the reviewer's comment that the paragraph needs to be revised to make clear that we do not infer any relationship from non-significant correlations.*

L429: Why does a seasonal forecast model have lower signal-to-noise ratios?

*We thank the reviewer for raising this question. While we cannot define signal-to-noise in ERA5 in the same way (given that we have a single realisation), we will consider removing this sentence if we are not able to fully justify it, or, alternatively refer to the signal-to-noise paradox.*

**Technical comments:**

There are quite a few typos, missing words, and instances of poor grammar throughout the paper. I will highlight a few examples here but there are far too many and I would recommend a more thorough edit and grammar check before re-submission.

*We thank the anonymous reviewer for carefully reading the manuscript and for spotting grammatical errors and typos. In the revised version of the manuscript, we will make sure to improve this aspect of the manuscript and address of the highlighted issues.*

---

## Author Comment (AC2)

**Final response to Anonymous Referee #2**

*We thank the Anonymous reviewer #2 for their positive and encouraging review. Below we explain in detail how we intend to address the reviewer's comments during the revision process. Our replies are highlighted in italics.*

**General comments**

This paper represents a crucial effort in identifying potential shortcomings in model representation of recent Greenland blocking trends and is therefore an important contribution to the literature. I found the authors' methods to be well-suited to the objectives of the paper and thought that the conclusions were mostly well-founded. My main critique is that more careful consideration and in-depth discussion of the implications of the authors' choice of blocking detection method are warranted. Given that much of the work demonstrating a positive trend in Greenland blocking – including the Preece et al. 2023 hypothesis that this work tested – was based on a field-departure-based blocking index, how might the application of a reversal-based detection method in this study impact the results presented herein and how they compare to previous works?

*We thank the reviewer for their positive outlook on the manuscript. Considering that the reviewer rises a similar point to that by anonymous reviewer #1, i.e. the possible dependence of the results on the chosen metric, we plan to address this point in the revised version of the manuscript by including the GBI index as defined in Preece et al. (2023), i.e. the area averaged 500 hPa geopotential height within the domain of 60-80°N and 20-80°W. We might expect some difference in trends, with the reversal index being strong, but the overall findings should be similar (Luu et al. 2024). However, this addition should shed light on whether the conclusions we draw from this analysis are robust when different indices are employed.*

*Luu, L.N., Hanna, E., de Alwis Pitts, D. et al. Greenland summer blocking characteristics: an evaluation of a high-resolution multi-model ensemble. Clim Dyn 62, 10503–10523 (2024). https://doi.org/10.1007/s00382-024-07453-2*

The authors' use of the deconstructed components of the reversal-based blocking index in their causal discovery method was particularly novel and interesting; however, I do wonder if too much emphasis was placed on the GHGN criterion in distinguishing between anticyclonic conditions and Greenland blocking. For example, L441 states, "Thus, in ERA5 our findings generally support the first part of Preece23's hypothesis, showing that T2M-Arctic and May snow cover may influence MSLP over North America, which in turn favours pressure highs over Greenland (GHGS>0). However, this does not consistently lead to blocking, as MSLP also contributes to GHGN>0, reducing the likelihood of blocking." I'm not sure that this distinction is quite so definitive. For example, Tyrlis et al. (2021) argue that high-latitude blocks such as those that impact Greenland are distinct in that they shift the jet stream to the south and, consequently, requiring strong westerly flow to the north may not be appropriate. They argue that the poleward geopotential height gradient criterion should be relaxed to 0 m per degree latitude for locations north of 60∘N latitude. How might this argument impact the interpretation of the seemingly contradictory links with the GHGN index revealed by the authors?

*We thank the reviewer for their insightful comment. In the revised version of the manuscript, we will reassess our interpretation of the GHGN behavior considering the Tyrlis et al. (2021) results. We will also explore the possibility to apply their detection method, even if this will not affect the specific results obtained with GHGN and GGI. More in general, checking for the robustness of the results depending on different blocking indices (following the previous point), will help us understand whether blocking an absolute-field index like the GBI used by Preece et al. (that do not depend on the*

*reversal-based detection) shows consistent results with GGI or GHGS, or both. If consistent results are found with GHGS, but not GHGN, this would be further evidence that GHGS may be a better proxy for blocking over Greenland than GHGN, and that GHGN may not be as important in this region to define blocking, as suggested in Tyrlis et al. (2021).*

**Specific comments**

*We thank the anonymous reviewer for their thorough read of our manuscript. Below we reply to all comments, other than those highlighting a typo or a suggestion to rephrase a sentence (which will be dealt with in the point-by-point reply following the revision).*

L184: What is the reason for extending the domain as far east as the prime meridian? Why start the southern bound of the domain at 67N?

*The domain of the GBI region is identified based on where a significant increase in GBI index is detected (Fig. 1a, currently described in lines 253-255). In the revised version of the manuscript, we will make sure that this key information is better highlighted when the definition of the domain first appears.*

L185: The Greenland Blocking Index, or GBI, has already been well established with a specific definition of the average 500 hPa height within the domain of 60-80N and 20-80W. I strongly suggest that the name here is altered to distinguish the index defined herein from the established GBI. Perhaps something as simple as the reversal-based Greenland blocking index (rGBI).

*As state earlier, we plan to address the effect of different blocking indices as highlighted by the reviewers. Thus, following the reviewer's suggestion, we will name identify the reversal-based Greenland blocking index as "rGBI" in the revised version of the manuscript.*

L322: Here you note that 33% of the GGI>1 s.d. T2m-G fall above the 90th climatological quantile in SEAS5.1-03; however, Figure 3f indicates 35.3% fall above the 90th climatological quantile. Which is correct?

*We thank the anonymous reviewer for pointing out this discrepancy. There is indeed a mistake in the text, and we will update the revised version of the manuscript with the correct percentage (35%).*

Figure 3: The meaning of the red shading and the text annotations in panels (e) and (f) should be noted in the figure caption.

*Following the reviewer's suggestion, we will add the meaning of the red shading and the text annotations in panels (e) and (f) in the caption of Fig. 3.*

Figure 4: The caption title is a bit confusing. Do the time series in the right column show the 11-year running mean of index values or an 11-year moving window trend of monthly-mean index values? The units at the top of each plot would suggest the latter, but the caption title suggests the former.

*Figures 4i-l show the 11-year moving window trend of monthly-mean index values. In the revised version of the manuscript we will revised the caption title to avoid confusion.*

L408: GBI is repeated here. Should one of these be GGI?

*We thank the anonymous reviewer for spotting this mistake, the sentence should indeed read "GBI and GGI". We will correct this mistake in the revised version of the manuscript.*

L418-421:  The stationary wave response should increase as the background westerly flow weakens due to Arctic amplification. This could explain why the relationship with NA snow cover anomalies is stronger in the ERA5-81 record.

*We thank the anonymous reviewer for point out this relevant literature, which we will include in the discussion of our results in the revised version of the manuscript.*

L462: FDR has not been defined

*We thank the anonymous reviewer for point out this missing information. FDR stands for "false discovery rate", which is described in lines 219-220. However, the definition of the acronym was missing. We will correct this discrepancy in the revised version of the manuscript.*

Figure 6: More explanation is needed in the figure caption. What is the meaning of the numbers on the linkage arrows? Why do some connecting lines not include an arrow head? Why are there two color bars included at the bottom of the figure (i.e., what does each bar correspond to?) I see that this information is given on L232, but it would be helpful to have it in the caption as well.

*We will revise the caption of Fig. 6 following the reviewer's suggestion.*

Figure 7a: Where does this CEN diagram come from? Is this based on the analysis summarized in Figure 6? If so, why is the lag-0 linkage between MSLP-NAm and GGI positive?

*The definition of the CEN in Fig. 7a is currently described in lines 521-523: "To provide a fair comparison between ERA5-81 and SEAS5.1-03, we adopt the concept of causal inference (see Methods Section 2.3). Based on the results shown in Fig. 6, we now assume that the selection of causal links shown in Fig. 7a is found in both ERA5-81 and SEAS5.1-03, and also for GHGS and GHGN indices.". We agree with the reviewer's comment that this panel requires more explanation. Here, we construct the CEN network by imposing a sub-selection of all the links that PCMCI detects in Fig. 6a and 6b. Links are sub-selected to best represent the Preece et al. (2023) hypothesis and to balance between links found in ERA5 and those found in SEAS5.1. Another difference between Fig. 6a and 7a is that in Fig. 7a the maximum lag used is -1. The sign of the MSLP-NAm -> GGI link at lag 0 changes from negative (Fig. 6b) to positive (Fig. 7a) because Fig. 7a show the CEN of ERA5, while Fig. 6b shows the CEN network for SEAS5.1. However, changes in sign between Fig. 6a and 7a (both showing ERA5 CENs) are also seen. For example, the T2m-Arctiv -> GGI link goes from positive in Fig. 6a to negative in Fig 7a. This does not however represent a contradiction since the lag of the link in Fig. 6a is -2, while in Fig. 7a we restrict it to -1 (for consistency with SEAS5.1).*

---

## Author Response (AR1)

*We thank the Anonymous reviewer #1 and #2 for their insightful and encouraging review. Below we explain in detail how we have addressed the reviewer's comments. Our replies are highlighted in italics.*

**Point-by-point response – Anonymous Referee #1**

**Major comments**

1. Observations and reanalysis are repeatedly conflated in the paper. Reanalysis is still a model-derived product, and its snow cover is biased (e.g. Mudryk et al 2015,Mortimer et al 2020) when comparing to in-situ and observation-derived gridded products. This makes me wonder how dependent the results in this paper are on the use of ERA5 as 'observations', and I'd recommend first that the authors are more careful about their use of the word observations, and second that some discussion around how ERA5's biases could be impacting the results. I also wonder whether other reanalysis products would be able to reproduce the same causality? Or whether a different metric for GB would yield similar results, both for causality and for how unusual the reanalysis trend is. I wonder as well where there is a state-dependence and how that might come into play, for example a non-linearity when future snow cover over North America is much lower on average?

*We agree that – despite being a common terminology in climate modelling – using the word "observations" when reanalysis data are used can generate confusion. Therefore, we correctly now refer to "reanalysis" in the revised version of the manuscript.*
*Following the reviewer's suggestion, we have added to our analysis the absolute Greenland Blocking Index developed by Hanna et al. (2016) - defined as HA16 index in the text - as the area averaged 500 hPa geopotential height within the domain of 60-80°N and 20-80°W. This is the same index described in Preece et al. 2023 to our analysis. Using the HA16 index leads to qualitatively similar results, thus showing that our analysis does not depend on the type of blocking index used. These complementary results are described in multiple sections of the text*

- *L237-239:* Similar results can be obtained when using different blocking indices, as the HA16 and TY21 indices. Despite minor differences inherently associated with their definitions, both indices display the same qualitative trends as seen in Fig. 1c (Fig. S2-3 in the Supplementary Material)
- *L266:* We also check the trends obtained with the HA16 index, which shows qualitatively similar results (see Fig. S6 in the Supplementary Material).
- *L308-310:* The same Figures but for GGI SEAS5.1-05, for SEAS5.1-03 GHGS and GHGN, as well as for the HA16 index show consistent results, even though for positive GHGN values the sign of the Z500 and T2m anomalies is inverted (see Supplementary Material, from S7 to S10).
- *L439-441* Using the HA16 index leads to qualitatively similar results for both ERA5 and SEAS5.1 (see Fig. S13 in the Supplementary Material), even though, as expected, some stronger relationship emerges for ERA5.
- *L 487-488:* In general, similar causal links are also detected when GHGS, GHGN and the HA16 index are analyzed (Figs. S14-15 in Supplementary Material)
- *L552-553:* Using the HA16 index leads to qualitatively similar results (see Fig. S16 in the Supplementary Material).

*Concerning the other main point, we agree with the anonymous reviewer that the snow – GBI mechanism may display a non-linear behaviour with a decrease of snow cover over North America, which is already partially evident in the difference observed between the relationship in ERA5-81 and ERA5-40. We tackle this point together with potential dependence on tropical variability in the next point.*

2. Despite the title, quite a lot more time is spent on the idea that there is a forced positive trend in GB, driven by the Preece et al 2023 mechanism, rather than the idea that natural variability (in particular anything other than the AMV), or even a forced increase in variability, has caused the trend in reanalysis. Evidence from CMIP6 is that the forced trend is negative with a lot of variability super-imposed, and so even if the Preece et al 2023 mechanism is correct and is missing from models, it's not obvious to me that that means the models are wrong in the direction of their trend. Perhaps the forced trend for GB is not driven from the pole, but rather from the lower latitudes (on balance) and that's the source of the decline in future GB? I do agree, however, that a missing mechanism that increases GB variability on an interannual timescales could still be important for future Greenland melt, and I do think that the results here are useful science, I'm just not sure about the way it has been framed.

*We agree with the anonymous reviewer that, while our analysis suggests that the snow cover mechanisms is missing in SEAS5.1, and potentially also in CMIP models – which are not part of this study –, this does not rule out that other mechanisms are at play and that (natural) tropical variability could be influencing the observed trend. While we do not directly focus on tropical forcings in this manuscript, we have also checked how the impact of El Nino - measured via the Nino3.4 index - can affect the snow cover – GBI mechanism. This is done following Tian et al. (2024) (see their Fig. 6).*
*In a nutshell, years are ordered based on the value of the Snow-NAm coverage in May (Fig. R1b,d,f) or the JJA Nino3.4 index (Figs. R1a,c,e). After that, the Snow-NAm → GGI causal link is recalculated both for ERA5 and SEAS5.1 on an 11-year moving window. Then we calculate the linear trend to see how the β value changes during years with high (or low) snow cover and El Niño (or La Niña) signature. For ERA5, results show that the Snow-NAm → GGI link is negative and stronger during La Niña years (Fig. R1a) and during years with higher snow cover in May (Fig. R1b). SEAS5.1 generally fails to reproduce this signal, with the exception of SEAS5.1-05, where the Snow-NAm → GGI link strengthens and is negative for high snow cover (Fig. R1f).*

*Moreover, we also checked whether including the Nino3.4 index in the network would show any significant links (Fig. R2). No links to or from Nino3.4 are shown but this does not exclude that the background state of ENSO influences the CEN (as shown in Fig. R1) nor that nino3.4 may have an indirect link to GGI via variables that are not included in this study.*

*Of course, having ruled out the impact of ENSO on the GBI does not imply that any other specific tropical forced pattern can impact the blocking over Greenland, but in absence of any other evident mechanism we would prefer to keep our focus on local phenomena as the snow cover role.*

*Tian, Y., Giaquinto, D., Di Capua, G. et al.: Historical changes in the Causal Effect Networks of compound hot and dry extremes in central Europe. Commun Earth Environ 5, 764 (2024).*

*https://doi.org/10.1038/s43247-024-01934-2*

[Figure]

*Fig. R1. Panel (a): dependence of the β value of the Snow-NAm → GGI link at monthly time scale on the ENSO interannual variability (JJA nino3.4 index) for ERA5. Panel (b): Same as for panel (a) but for the dependence on Snow-NAm in May. Panel (c): Same as for panel (a) but for SEAS5.1-03. The linear regression is calculated from the ensemble mean of each year (for each year, 1000 beta values are calculated with bootstrapping). Panel (d): Same as for panel (c) but for the dependence on Snow-NAm in May. Panels € and (f): same as for panels (c) and (d) but for SEAS5.1-05.*

[Figure]

*Fig. R2. CEN as in Fig. 6a but adding the nino3.4 index.*

3. The intro and the conclusions are both long and meandering at times between forcing of GB between the tropics, midlatitudes and poles, and between climate models and observations. Please consider re-writing to make it clearer.

*Following the reviewer's comment, we have shortened and simplified both the introduction and discussion sections, thus improving the readability and clarity of the revised manuscript.*

4. I don't think using T2m-Arctic as an indicator for Arctic amplification is sufficient. A difference between the Arctic and some mid-latitude band would probably be better, as a year with high T2m Arctic could also have high temperatures in general, i.e. T2m Arctic is highly correlated with T2m global. In general, I think the term Arctic amplification is used when the authors intend to say Arctic warming, so I'd recommend more careful wording.

*Following the suggestion of the anonymous reviewer, we now refer to Arctic warming instead of Arctic amplification in the revised version of the manuscript.*

**Minor comments**

L143: Is the mean of each month for the entire period removed from that month? The following sentence is obvious and need not be included.

*We have removed this sentence following the reviewer's suggestion.*

L150 Why isn't April one of the initialisations for SEAS5.1?

*We chose to analyse both 1st March and 1st May initialisation dates for SEAS5.1 to assess the dependency of the results on the length of the forecasts prior to the target season, e.g on the forecast lead time. It would be reasonable to assume that forecasts initialized on 1st April, may show a mixed signal of both the earlier and later initialised runs. However, since we do not detect significant differences between SEAS5.1-03 and SEAS5.1-05, it is reasonable to assume that an initialisation date in between would not diverge from the obtained results.*

L155: Everything after 'Liner correlation should be moved to the section 2.2

*Lines 155-158 have been moved to section 2.3 in the revised version of the manuscript,*
 - *L208-210: "*Here, linear correlation analysis and probability-trend estimation were conducted for all datasets mentioned. The correlation and causal inference analyses were performed for both SEAS5.1-03 and SEAS5.1-05, except for CENs. For causal discovery, only SEAS5.1-03 was used, as the absence of March and April data in SEAS5.1-03 violates PCMCI algorithm requirements.*"*

Figure 2: It's interesting that there's a reversal in the positions of ERA-40 and ERA-81 in terms of their percentile between GBI and GGI. The red lines do not look to be correlated in (c) and (d), as in Figure 1(c). Is there is a mistake in the plot or in the caption? Why is GHGS and GHGN written on panels (c) & (d)?

*We thank the reviewer for pointing out this mistake. We have corrected the figure text in the revised version of the manuscript.*

Figure 3: I wonder if a different plot of (b)-(a) would be helpful for visualising where ERA5 and SEAS5.1 differ.

*Following the reviewer's suggestions, we now provide the additional panels in Fig. 3 showing the difference between ERA5 and SEAS5.1-03 for both T2m and Z500 fields. These new panels are described in lines 293-295 "The difference between SEAS5.5-03 and ERA5 composites for both T2m and Z500 shows a tendency of the model to have higher Z500 anomalies over Greenland and the North Atlantic and higher T2m anomalies over Greenland and Northern Canada with respect to ERA5 (Figs. 3c-d).".*

Figure 4: (j) It's interesting that all the members are so tightly constrained for Snow-NAm compared to other fields, and I wonder why that might be, and if it's showing a related issue, whereby the seasonal model is not simulating variability in snow cover properly?

*Being the seasonal forecast simulations initialised for a state as close as possible to observations, they tend to diverge with increasing forecast lead time. However, fields such as sea surface temperature (SST) or snow cover are characterized by larger inertia and have a slower variability than atmospheric fields such as T2m or Z500, so it is absolutely expected that they diverge from the initialization state more slowly. Therefore, both AMV (which is derived from SST) and snow cover time series in Figs. 4i and 4l show smaller spread around the average values when compared to T2m and MSLP fields. Moreover, for snow cover (Fig. 4i), only May is considered, and because these plots are obtained using SEAS5.1-05, the divergence from the initial state is minimal. Therefore, this behaviour should not highlight an underlying issue, but rather an expected behaviour of the seasonal forecast fields. We explain this point clearly in the revised version of the manuscript, in lines 373-375 "Note that both AMV and Snow-NAm SEAS5.1 indices are more tightly constrained with respect to are indices. This is due to the fact that SST or snow cover fields are characterized by larger inertia and have a slower variability than atmospheric fields, such as T2m or Z500, thus this behaviour is expected.".*

Paragraph L 396: non-significant correlations can't support a relationship, the only thing that's been shown there is that Arctic temp and GB are correlated.

*While we think it is worthwhile to report non-significant correlations (given that the significance is also affected by the shortness of the time series), we agree with the reviewer's comment that the paragraph needs to be revised to make clear that we do not infer any relationship from non-significant correlations. We do so in lines 397-401, "We first focus on the GBI index in relation to Preece23's hypothesis. In ERA5-81, positive (lag 0) correlations are found between both T2M-Arctic, MSLP-NAm and GBI (r ~ 0.3), while Snow-NAm (taken in May) shows a negative correlation with GBI (r ~ -0.2). The correlation between AMV and GBI is r ~ 0. However, only the correlation between T2M-Arctic and GBI is statistically significant (p < 0.05), in keep with the first part of the hypothesis linking Arctic warming to enhanced GBI (Fig. 5b).".*

L429: Why does a seasonal forecast model have lower signal-to-noise ratios?

*We thank the reviewer for raising this question. One of the most intriguing issues of the current seasonal climate modeling research is that current state of the art climate prediction systems tend to suffer from a "statistical" issue defined as Signal to Noise Paradox (SNP). Practically, this implies that ensemble mean forecast correlates better with the observation than with individual members of the forecast ensemble. A consequence is that predictable signal, which is associated with signal (the temporal variability of the ensemble mean) to noise (the ensemble spread) ratio, is often smaller than the one of the real world. For example, it is estimated that over the North Atlantic the predictable signal in the model is smaller by a factor of two than in observations. To better express this point, we rephrased at lines 420-422:*

*"SEAS5.1-03 generally weaker correlations, which are somehow expected given the usual small signal-to-noise ratio of prediction systems typical over the North Atlantic sector (Scaife and Smith, 2018)."*

*References:*

*Scaife, A.A., Smith, D. A signal-to-noise paradox in climate science. npj Clim Atmos Sci 1, 28 (2018). https://doi.org/10.1038/s41612-018-0038-4*

*Weisheimer, A., and Coauthors, 2024: The Signal-to-Noise Paradox in Climate Forecasts: Revisiting Our Understanding and Identifying Future Priorities. Bull. Amer. Meteor. Soc., 105, E651–E659, https://doi.org/10.1175/BAMS-D-24-0019.1.*

**Technical comments:**

L15 incorrect use of colon.
*Sentence has been revised (line 15).*

L21 climate runs -> climate model runs
*Revised as suggested.*

L29 ice melt -> ice sheet melt
*Revised as suggested.*

L100 of representing blocking -> to represent blocking
*Revised as suggested.*

L115 casual -> causal
*Revised as suggested.*

L162 their identification -> its identification
*Revised as suggested.*

L174 I think the use of 'condition' isn't the correct word, as those are the three equations after L180, what is here is a definition.
*Revised as suggested.*

L225 Need to define s.d.
*Revised as suggested.*

Section 2.3 doesn't need its own subsection check

*Subsection 2.4 has been removed and the text is now part of subsection 2.3.*

L251 yearly – seasonal
*Revised as suggested.*

L253 variability -> spatial variability
*Revised as suggested.*

Figure 2 has (a) & (b) have (9095) squished together.

*Adjusted*

L339-340: 'above its own 1 s.d.' is awkward wordings

*Revised as suggested.*

L279 & elsewhere: running mean -> running mean trend.

*Following the suggestion of Reviewer #2, "running mean trend" has been substituted with "moving window trend".*

L 475 'higher' -> lower? As the sign is negative? Same with snow anomalies below.

*We checked lines 476-440*

*"MSLP-NAm shows only one incoming positive (β = 0.15) link from Snow-NAm at lag −2, meaning that higher than average Snow-NAm values in AMJ are followed by higher MSLP anomalies over eastern North America, also in keeping with Preece23. Snow-NAm shows three incoming links: one positive link from MSLP-NAm at lag −1 (β = 0.15), and two negative links, one from T2m-Arctic at lag −1 (β = -0.3) and one from AMV at lag −2 from AMV (β = -0.15).",*

*This refers to the description of Fig. 6a, and to our understanding is correct: the link from Snow-NAm to MSLP-NAm is indeed positive, while the link from MSLP-NAm to Snow-NAm is also positive. On the other hand, the links from AMV and T2m-Acrtics towards Snow-NAm are negative.*

L509: keep -> keeping
*Revised as suggested.*

L530 – 545: issues with font, I think arising each time 'beta' is written.
*The font has been revised)*

**Point-by-point response – review #2**

**General comments**

This paper represents a crucial effort in identifying potential shortcomings in model representation of recent Greenland blocking trends and is therefore an important contribution to the literature. I found the authors' methods to be well-suited to the objectives of the paper and thought that the conclusions were mostly well-founded. My main critique is that more careful consideration and in-depth discussion of the implications of the authors' choice of blocking detection method are warranted. Given that much of the work demonstrating a positive trend in Greenland blocking – including the Preece et al. 2023 hypothesis that this work tested – was based on a field-departure-based blocking index, how might the application of a reversal-based detection method in this study impact the results presented herein and how they compare to previous works?

*We thank the reviewer for their positive outlook on the manuscript. The presence of a trend in summer Greenland Blocking has been already documented also in a reversal-based index, as in Davini and D'Andrea (2020), so this is why we started looking at this family of indices. However, considering that the reviewer rises a similar point to that by anonymous reviewer #1, we appreciate that a clarification is necessary and thus we included the Greenland Blocking Index developed by Hanna et al. (2016) - marked as HA16 index hereafter and in the text - in our analysis. Using the HA16 index leads to qualitatively similar results, thus showing that our analysis does not depend on the type of blocking index used, and actually providing further strength to the conclusions drawn. These complementary results are described in multiple sections of the text:*

- *L237-239:* Similar results can be obtained when using different blocking indices, as the HA16 and TY21 indices. Despite minor differences inherently associated with their definitions, both indices display the same qualitative trends as seen in Fig. 1c (Fig. S2-3 in the Supplementary Material)
- *L266:* We also check the trends obtained with the HA16 index, which shows qualitatively similar results (see Fig. S6 in the Supplementary Material).
- *L308-310:* The same Figures but for GGI SEAS5.1-05, for SEAS5.1-03 GHGS and GHGN, as well as for the HA16 index show consistent results, even though for positive GHGN values the sign of the Z500 and T2m anomalies is inverted (see Supplementary Material, from S7 to S10).
- *L439-441* Using the HA16 index leads to qualitatively similar results for both ERA5 and SEAS5.1 (see Fig. S13 in the Supplementary Material), even though, as expected, some stronger relationship emerges for ERA5.
- *L 487-488:* In general, similar causal links are also detected when GHGS, GHGN and the HA16 index are analyzed (Figs. S14-15 in Supplementary Material)
- *L552-553:* Using the HA16 index leads to qualitatively similar results (see Fig. S16 in the Supplementary Material).

The authors' use of the deconstructed components of the reversal-based blocking index in their causal discovery method was particularly novel and interesting; however, I do wonder if too much emphasis was placed on the GHGN criterion in distinguishing between anticyclonic conditions and Greenland blocking. For example, L441 states, "Thus, in ERA5 our findings generally support the first part of Preece23's hypothesis, showing that T2M-Arctic and May snow cover may influence MSLP over North America, which in turn favours pressure highs over Greenland (GHGS>0). However, this does not consistently lead to blocking, as MSLP also contributes to GHGN>0, reducing the likelihood of blocking." I'm not sure that this distinction is quite so definitive. For example, Tyrlis et al. (2021) argue that high-latitude blocks such as those that impact Greenland are distinct in that they shift the jet stream to the south and, consequently, requiring strong westerly flow to the north may not be appropriate. They argue that the poleward geopotential height gradient criterion should be relaxed to 0 m per degree latitude for locations north of 60∘N latitude. How might this argument impact the interpretation of the seemingly contradictory links with the GHGN index revealed by the authors?

*We thank the reviewer for their insightful comment. In the revised version of the manuscript, we included some selected analysis making use of the criteria of Tyrlis et al 2021, defined as TY21 hereafter and in the text. Differences between the TY21 index and the one used in the current analysis are however minimal, as shown by the timeseries of the indices now reported in the Figure S3, which shows that the Pearson correlation is 0.82. Indeed, the two indices share the main GHGS conditions and even if the GHGN is loosen to 0 deg/m still detect mostly the same structure. We therefore assume that for the overall assessment, from the qualitative point of view, the two indices are somehow coincident.*

**Specific comments**

L38: Add a comma after "pattern"
*Revised as suggested.*

L43: Add a closing en dash between the in-text citation and "seem"
*This sentence has been removed in the revised version of the manuscript.*

L47: I think this would read clearer as "Greenland blocking is a large-scale atmospheric high-pressure, low-vorticity system located over Greenland that is associated with the negative phase of the North Atlantic Oscillation (Woollings and Hoskins, 2008).
*Revised as suggested.*

L71: I suggest rewriting this as "There is accumulating evidence that the frequency of summer Greenland blocking has increased over the last two decades…"
*Revised as suggested.*

L79: Rewrite as, "leaving open the possibility that the increase is a consequence of natural variability."
*Revised as suggested.*

L92: replace "contribute to inhibiting" with "inhibit"
*Revised as suggested.*

L100: replace "ability of representing" with "representation of"
*Revised as suggested.*

L113: replace "identifying" with "identify"
*Revised as suggested.*

L116: replace "identifying" with "identify"
*Revised as suggested.*

L152: should "till" be "until"?
*Revised as suggested.*

L184: What is the reason for extending the domain as far east as the prime meridian? Why start the southern bound of the domain at 67N?

*The choice of the GBI domain is based on the region where the blocking index shows a significant positive trend. We explain this in the revised version of the manuscript in lines 150-151: "Note that the GBI region is identified based on where a significant increase in the blocking index is detected (Fig. 1a)".*

L185: The Greenland Blocking Index, or GBI, has already been well established with a specific definition of the average 500 hPa height within the domain of 60-80N and 20-80W. I strongly suggest that the name here is altered to distinguish the index defined herein from the established GBI. Perhaps something as simple as the reversal-based Greenland blocking index (rGBI).

*We thank the reviewer for its suggestion, but after considering to rename our GBI index as "rGBI", we have decided to stick to GBI and refer to the the average 500 hPa height within the domain of 60-80N and 20-80W as the HA16 index. While the HA16 index is often used to describe summer blocking over Greenland, technically speaking an atmospheric blocking is not defined simply by the value of geopotential height measured at Z500. Atmospheric blocking is a specific circulation pattern characterized by an isolated high pressure system capable of diverting or blocking the migrating cyclones. Literature established blocking indices are usually divided into anomaly-based (which*

*detects anomalies of geopotential height) or gradient-reversal based (as the one currently adopted in the manuscript), or a combination of those, and they are applied on a latitudinal band or on 2-D domain. Despite its value for the specific task, the HA16 index is a punctual metric which does not fit either into the anomaly-based or gradient-reversal based indices. We therefore prefer to avoid using it as the reference for our Greenland Blocking Index.*

L255: replace "irrespectively" with "irrespective"
*Revised as suggested.*

L322: Here you note that 33% of the GGI>1 s.d. T2m-G fall above the 90[th] climatological quantile in SEAS5.1-03; however, Figure 3f indicates 35.3% fall above the 90[th] climatological quantile. Which is correct?

*We thank the anonymous reviewer for pointing out this discrepancy. We have corrected this mistake in the text, and we have updated the revised version of the manuscript with the correct percentage (35%), line 312.*

Figure 3: The meaning of the red shading and the text annotations in panels (e) and (f) should be noted in the figure caption.

*Following the reviewer's suggestion, we have added the meaning of the red shading and the text annotations in panels (e) and (f) in the caption of Fig. 3, line 298-300 "Red shading in the plot highlights the position of the climatological 50th and 90th percentiles, while the numbers below the percentiles show the percentage of T2m-G which is above the climatological percentile in the sub-selected PDF.".*

L347: replace "months is summer" with "summer months"
*Revised as suggested.*

L371: I believe AVM should be AMV here
*Revised as suggested.*

Figure 4: The caption title is a bit confusing. Do the time series in the right column show the 11-year running mean of index values or an 11-year moving window trend of monthly-mean index values? The units at the top of each plot would suggest the latter, but the caption title suggests the former.

*Figures 4i-l show the 11-year moving window trend of monthly-mean index values. We have revised the caption title and text to avoid confusion (lines 377-384).*

L408: GBI is repeated here. Should one of these be GGI?

*We thank the anonymous reviewer for spotting this mistake, the sentence should indeed read "GBI and GGI". We have corrected this mistake in the revised version of the manuscript (line 407).*

L418-421: The stationary wave response should increase as the background westerly flow weakens due to Arctic amplification. This could explain why the relationship with NA snow cover anomalies is stronger in the ERA5-81 record.

Hoskins, B., & Woollings, T. (2015). Persistent Extratropical Regimes and Climate Extremes. *Current Climate Change Reports*, *1*(3), 115–124. https://doi.org/10.1007/s40641-015-0020-8

Coumou, D., Di Capua, G., Vavrus, S., Wang, L., & Wang, S. (2018). The influence of Arctic amplification on mid-latitude summer circulation. *Nature Communications*, *9*(1), 2959. https://doi.org/10.1038/s41467-018-05256-8

*We thank the anonymous reviewer for pointing out this relevant literature, and we included this in the discussion section in order to improve the final summary of our findings. The following sentence is reported at L585-590:*

*Our correlation and causal analyses indicate a robust relationship in ERA5, where high Arctic temperatures lead to early snow cover depletion in May, which can subsequently cause a low-pressure anomaly over North America (MSLP-NAm), consistent with Preece23 (Figs. 5, 6 and 7). The identified relationships are stronger over the more recent time window 1981-2023 rather than the full reanalysis period 1940-2023: this is consistent with the larger impact of Arctic warming in the recent years, which could have weakened the background westerly flows and this favoured stronger stationary wave persistence (Hoskins and Woollings, 2015; Coumou et al, 2018).*

L462: FDR has not been defined

*We thank the anonymous reviewer for pointing out this missing information. FDR stands for "false discovery rate", which is described in lines 219-220. However, the definition of the acronym was missing. We have added the definition of the acronym in the revised version of the manuscript (line 196).*

L475: I believe this should be eastern US, not western US

*We have corrected this mistake, now referring to eastern North America(line 478).*

Figure 6: More explanation is needed in the figure caption. What is the meaning of the numbers on the linkage arrows? Why do some connecting lines not include an arrow head? Why are there two color bars included at the bottom of the figure (i.e., what does each bar correspond to?) I see that this information is given on L232, but it would be helpful to have it in the caption as well.

*We have revised the caption of Fig. 6 following the reviewer's suggestion: "In each CEN, actors are represented by a node in the network, while lagged causal relationships are represented by directed arrows, showing the direction of causality. Lag 0 links are shown as straight arrows without an edge. The colour of the arrows (nodes) shows the strength of the causal effect, the β value (auto β value), while the numbers on the arrows show the lag at which the causal link is detected.", lines 522-527.*

L530: replace "snow cover on North America" with "North American snow cover"
*Revised as suggested.*

Figure 7a: Where does this CEN diagram come from? Is this based on the analysis summarized in Figure 6? If so, why is the lag-0 linkage between MSLP-NAm and GGI positive?

*We agree with the reviewer's comment that this panel requires more explanation. We have revised the description of this panel in lines 529-535, "To provide a fair comparison between ERA5-81 and SEAS5.1-03, we adopt the concept of causal inference (see Methods Section 2.3). Based on the results shown in Fig. 6, we now assume that the selection of causal links shown in Fig. 7a is found in both ERA5-81 and SEAS5.1-03, and also for GHGS and GHGN indices. Here, we construct the CEN network by imposing a sub-selection of all the links that PCMCI detects in Fig. 6a and 6b. Links are sub-*

*selected to best represent the Preece et al. (2023) hypothesis and to balance between links found in ERA5 and those found in SEAS5.1. Note that in Fig. 7a the maximum lag used is -1. Moreover, while in Fig. 6b the sign of the SEAS5.1 MSLP-NAm → GGI link (lag 0) is negative, in ERA5 this link shows a positive sign (Fig. 7a)."*

---

## Author Response (AR2)

We thank the Anonymous reviewer for their last comment. We have addressed all minor comments as listed below. Our replies are highlighted in italics. We have also corrected Figure 6 as we had plotted partial correlation instead of the beta coefficient.

L35: replace "community" with "community's"
> *done*

L38: perhaps "…extreme melt in recent years"
> *extreme melt **years** in recent years.*

L42: I would delete "multiple"
> *done*

L69: I would replace "Despite" with "While".
> *done*

L73: Replace "until" with "through"
> *done*

L93: Replace "we aim at investigating the Greenland" with "we investigate Greenland…"
> *done*

L321: It is not clear to me what "monthly" is referring to here (Monthly time scales?, Causal links identified using monthly-mean time series?). Perhaps it could simply be deleted?
> *Deleted "monthly"*

L385: Replace "has" with "have"
> *Done.*

L602: Should "is" be "in"?
> *Yes, done.*